# Losing dimensions: Geometric memorization in generative diffusion

## Abstract

Generative diffusion processes are state-of-the-art machine learning models deeply connected with fundamental concepts in statistical physics. Depending on the dataset size and the capacity of the network, their behavior is known to transition from an associative memory regime to a generalization phase in a phenomenon that has been described as a glassy phase transition. Here, using statistical physics techniques, we extend the theory of memorization in generative diffusion to manifold-supported data. Our theoretical and experimental findings indicate that different tangent subspaces are lost due to memorization effects at different critical times and dataset sizes, which depend on the local variance of the data along their directions. Perhaps counterintuitively, we find that, under some conditions, subspaces of higher variance are lost first due to memorization effects. This leads to a selective loss of dimensionality where some prominent features of the data are memorized without a full collapse on any individual training point. We validate our theory with a comprehensive set of experiments on networks trained both in image datasets and on linear manifolds, which result in a remarkable qualitative agreement with the theoretical predictions.

## 1 Introduction

Generative diffusion models (Sohl-Dickstein et al., 2015) have achieved spectacular performance in image (Ho et al., 2020; Song and Ermon, 2019; S. et al., 2021) and video (Ho et al., 2022; Singer et al., 2022; Blattmann et al., 2023; B. et al., 2024b) generation and currently form the backbone of most state-of-the-art image generation software (Betker et al., 2023; Esser et al., 2024). The defining feature of these models is their remarkable ability to generalize on complex high-dimensional data distributions. However, diffusion models are known to be capable of fully memorizing the training set in the low-data regime (Somepalli et al., 2023a;b; Kadkhodaie et al., 2024), and in this regime have been shown (Ambrogioni, 2024; Hoover et al., 2023) to be mathematically equivalent to Dense Associative Memory networks, which are modern variants of the celebrated Hopfield model admitting very large memory storage capacity (Krotov and Hopfield, 2016). The ability of the models to memorize has widespread societal implications, as in case of memorization their adoption would likely violate existing copyright laws. Therefore, understanding the factors that lead to generalization and memorization has great practical and theoretical value and drives future developments. It is often conjectured that natural images and other high-dimensional natural data have most of their variability confined on a relatively small sub-space of the ambient space of all possible pixel-values (Peyré, 2009; Fefferman et al., 2016). This *latent manifold* charts the space of possible images, which is embedded in a significantly larger space of meaningless configurations of pixels. The dimensionality of the latent manifold provides an estimate of the richness of generalizations since it quantifies the number of (linearly independent) ways in which an image can be altered while remaining in the space of possible generations. From this geometric perspective, a network that fully memorized the training set corresponds to a zero-dimensional latent manifold (i.e., a collection of points), because the individual "memories" cannot be altered without leaving the space of possible generated images. Generative diffusion models have a rich mathematical structure that closely mirrors systems that have been heavily studied in statistical physics (Montanari,

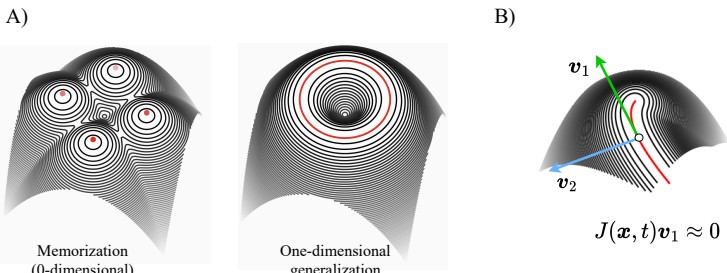

Figure 1: Visualization of the latent manifold of a diffusion model. The contour lines denote the log-probability (i.e. the (negative) 'energy'). The manifold of fixed points is drawn as a red line. A) Manifolds corresponding to memorization and one-dimensional generalization. B) Tangent and orthogonal singular vectors of the score.

2023; Raya and Ambrogioni, 2024; B. et al., 2024a; Biroli and Mézard, 2023; Ambrogioni, 2023). Recent works have shown that the transition from generalization to memorization in diffusion models is the result of a glass phase transition in the underlying energy function (B. et al., 2024a; Ambrogioni, 2023), which turns generative diffusion in a form of Dense Associative Memory network (Ambrogioni, 2024; Hoover et al., 2023). When the data is supported on a latent manifold, it is then natural to ask if all dimensions are lost at once due to memorization, or if instead they are lost gradually in separate transition events. In this paper, we refer to this intermediate memorization phenomenon as *geometric memorization* (Ross et al., 2024).

## 2 RELATED WORK

Diffusion Models (DMs) are the current state-of-the-art generative models in several domains, and works such as (De Bortoli, 2022; Pidstrigach, 2022) show that they are capable of learning the manifold structure of the data. There are several methods that, given a trained DM, can then estimate the Local Intrinsic Dimensionality (LID) at individual datapoints, defined as the dimension of the manifold in the component corresponding to the datapoint (Stanczuk et al., 2022; Horvat and Pfister, 2024; Kamkari et al., 2024b). Our analysis makes use of the method from (Stanczuk et al., 2022) to extend the analysis of manifold learning by DMs. The idea that memories can be detected as datapoints with lower LID is formalized in (Kamkari et al., 2024a). While our work shares similar conclusions, our focus is on the interplay between memorization and generalization for diffusion models, and provide a theoretical explanation of such phenomenon. Furthermore, (Kadkhodaie et al., 2024) show, both theoretically and empirically, how generalization arises in diffusion models as a function of the number of datapoints as well as the way the manifold dimension varies along the diffusion trajectory. While their findings are related to our results, our analysis differs as we analyze the nonparametric empirical score and we provide for a statistical physics perspective similar to the work of (Biroli and Mézard, 2023) on mixture of Gaussian data. This differs from (W. et al., 2024) which considers parameterized low-rank score functions. The advantage of the non-parametric empirical formulation is that it provides insight on the large capacity regime, which can cast insight on the behavior of modern large scale architectures. Regarding the usage of statistical mechanics, our work takes inspiration from the analysis conducted by (B. et al., 2024a; Raya and Ambrogioni, 2024; Ambrogioni, 2023) to describe the backward process of DMs as a series of phase transitions similar to those ones occurring in disordered systems.

## 3 GENERATIVE DIFFUSION MODELS

We will consider a Brownian process where an ensemble of "particles" $\boldsymbol{x}_0$ starts from an initial distribution $p_0(\boldsymbol{x})$ and then evolves through the stochastic equation

$$\mathrm{d}\boldsymbol{x}_t = \mathrm{d}\boldsymbol{Z}_t \tag{1}$$

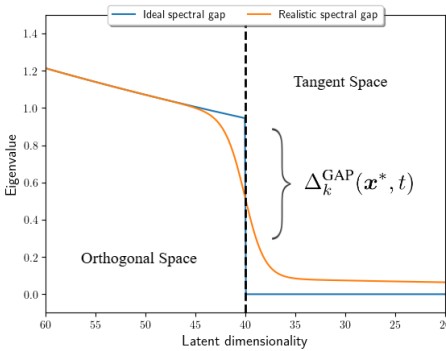
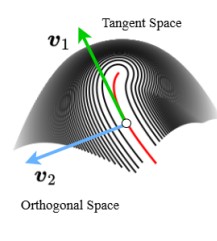

Figure 2: Illustration of the gaps in the singular values of the Jacobian of the score function in the presence of a latent manifold. The small values correspond to the tangent manifold (possible generations) while the high values correspond to the orthogonal manifold (forbidden generations). The singular values determine the steepness of the potential well along each eigen-direction.

where $\mathrm{d}\boldsymbol{Z}_t$ is a standard Wiener process. In generative modeling applications, $\boldsymbol{x}_0$ is usually chosen to be the distribution of data such as natural images. Eq. (1) is "solved" by the following formal expression: $p(\boldsymbol{x}_t, t) = \mathbb{E}_{\boldsymbol{x}_0 \sim p_0} \left[ \frac{1}{(2\pi t)^{d/2}} e^{-\frac{\|\boldsymbol{x}_t - \boldsymbol{x}_0\|_2^2}{2t}} \right]$ , where $d$ is the dimensionality of the ambient space. The *target distribution* $p_0(\boldsymbol{x})$ can be recovered by using a reversed diffusion process (Anderson, 1982). At large time $t_f$, we start from a sample $\boldsymbol{x}_{t_f} \sim \mathcal{N}(0, t_f I_d)$, and let it evolve through the backward process defined by

$$\mathrm{d}\boldsymbol{x}_t = -\nabla_{\boldsymbol{x}} \log p_t(\boldsymbol{x}_t) \mathrm{d}t + \mathrm{d}\boldsymbol{Z}_t \tag{2}$$

backward from $t_f$ to $t_0 = 0$. In the machine learning literature, the term $s(\boldsymbol{x}, t) = \nabla_{\boldsymbol{x}} \log p_t(\boldsymbol{x})$ is commonly referred to as the score function. These formulas are given according to what is known as the variance-exploding framework in the generative diffusion literature. However, all results can be ported directly to the variance-preserving (i.e. Ornstein–Uhlenbeck) case.

Given a training set $\{\boldsymbol{y}^1, \ldots, \boldsymbol{y}^N\}$ sampled from $p_0(\boldsymbol{x})$, we can obtain a neural approximation of the score function by training a noise-prediction network $\hat{\boldsymbol{\epsilon}}_{\boldsymbol{\theta}}(\mathbf{x}, t)$ (parameterized by $\boldsymbol{\theta}$) using the empirical denoising score matching objective (Hyvärinen and Dayan, 2005; Vincent, 2011; Ho et al., 2020). The network $\hat{\boldsymbol{\epsilon}}_{\boldsymbol{\theta}}(\boldsymbol{x}_t, t)$ is trained to predict the added noise $\boldsymbol{\epsilon}$ from a noisy state $\boldsymbol{x}_t = \boldsymbol{x}_0 + \sqrt{t}\boldsymbol{\epsilon}$. The estimated score is then given by $\hat{s}_{\boldsymbol{\theta}}(\boldsymbol{x}, t) = -\frac{\hat{\boldsymbol{\epsilon}}_{\boldsymbol{\theta}}(\boldsymbol{x}, t)}{\sqrt{t}}$ . Learning $\hat{\boldsymbol{\epsilon}}_{\boldsymbol{\theta}}$ instead of $\hat{s}_{\boldsymbol{\theta}}$ has the advantage of maintaining finite the output of the network for small $t \to 0$, where we known that the magnitude of the score becomes infinite outside of the support of the data.

## 4 LOCAL GEOMETRY AND LATENT MANIFOLDS

Here, we assume that the data distribution $p_0(\boldsymbol{x})$ is supported on a $d$-dimensional manifold $\mathcal{M}_0$ embedded in $\mathbb{R}^n$. We study the family of latent manifolds $\mathcal{M}_t$ for different values of the diffusion time $t$. When the distribution inside the manifold is uniform, these manifolds can be defined as sets of stable fixed-points of the score function:

$$\mathcal{M}_t = \{\boldsymbol{x}^* \mid s(\boldsymbol{x}^*, t) = \boldsymbol{0}, \text{ with } s(\boldsymbol{x}^*, t) \text{ n.s.d.}\} \tag{3}$$

where the negative semi-definiteness (n.s.d.) condition refers to the Jacobian of the score function at $\boldsymbol{x}^*$. Due to the Gaussian annulus theorem, during high-dimensional generative diffusion the samples "orbit" a shell around the manifold. However, while the manifold itself is not usually visited by the particles, its shape reflects the shape of the potential around it, which guides the generated trajectories. In order to understand the generative process, it is therefore important to study how the manifold emerges and changes during generation, and

how its shape depends on the distribution of the data and on the number of training samples. Its local geometry can be studied by analyzing the linearization of the score function around each point (Stanczuk et al., 2023). More precisely, for a given point $\boldsymbol{x}^*$ in $\mathcal{M}_t$, we can analyze the effect of adding a small perturbation vector $\boldsymbol{p}$ with magnitude in the order of $\sqrt{t}$:

$$s(\boldsymbol{x}^* + \boldsymbol{p}, t) \approx J(\boldsymbol{x}^*, t)\, \boldsymbol{p}\ , \tag{4}$$

where $J(\boldsymbol{x}^*, t)$ is the *smoothed Jacobian matrix* $J(\boldsymbol{x}, t)$, whose columns are defined as

$$\boldsymbol{J}_j(\boldsymbol{x}, t) = \left[ s(\boldsymbol{x} + \sqrt{t}\boldsymbol{e}_j, t) - s(\boldsymbol{x}, t) \right] / \sqrt{t}\ , \tag{5}$$

where $\boldsymbol{e}_j$ is a vector in an orthonormal basis set, which converges to the exact Jacobian of the score for $t \to 0$. This discretization in the definition has the advantage of only considering perturbations on the scale given by $\sqrt{t}$, since smaller perturbations cannot be resolved at the given noise level (Stanczuk et al., 2023). The effect of the linear perturbations can be expressed in terms of the singular values of $J(\boldsymbol{x}^*, t)$:

$$J(\boldsymbol{x}^*, t)\, \boldsymbol{p} = \sum_j \lambda_j(\boldsymbol{x}^*, t)\, \boldsymbol{v}_j\, (\boldsymbol{w}_j \cdot \boldsymbol{p})\ , \tag{6}$$

where the $\boldsymbol{w}_j$ ($\boldsymbol{v}_j$) is the $j$-th right (left) singular vector and $\lambda_j(\boldsymbol{x}^*, t)$ is its associated (non-negative) singular value. The right singular vectors $\boldsymbol{w}_j$ define a set of linearly independent perturbations of $\boldsymbol{x}^*$, while the scaled left singular vectors $-\lambda_j(\boldsymbol{x}^*, t)\boldsymbol{v}_j$ give a linearization of the score at the perturbed point. Generally, $\boldsymbol{v}_j$ is roughly aligned to $\boldsymbol{w}_j$, meaning that the tend to push the diffusing particle back towards the point on the manifold with a strength determined by $\lambda_j(\boldsymbol{x}^*, t)$. For $t \to 0$, the matrix $J(\boldsymbol{x}, t)$ becomes symmetric and the singular values correspond to the negative of the eigenvalues. We refer the set of singular values of $J(\boldsymbol{x}^*, t)$ as its *spectrum*. By analyzing these spectra, we can extract important information concerning the local geometry of $\mathcal{M}_t$. For a given $\boldsymbol{x}^*$, consider the set of left singular vectors that are orthogonal to $T_{\mathcal{M}_t}(\boldsymbol{x}^*)$. These vectors correspond to perturbations that move the particle orthogonally outside of the manifold, which are therefore associated with large singular values. On the other hand, perturbations that lie inside the tangent space correspond to a set of vanishing singular values since along these directions the particles can diffuse almost freely. This is visualized geometrically in Fig. 2 B. From these considerations, it follows the dimensionality of $T_{\mathcal{M}_t}$ can be estimated as the dimensionality of the right singular space of $J(\boldsymbol{x}^*, t)$ with 0 singular value. The drop between non-zero and zero singular values can be detected as a discontinuity (i.e. a gap) in its spectrum (see Fig. 2) (Stanczuk et al., 2023). In real datasets, the target distribution is generally not uniform when restricted to the manifold. In this case, the latent manifold cannot be defined as a set of fixed-points, we can still define $\mathcal{M}_t$ by annealing the target distribution into a uniform distribution defined over its support. Aside from definitional issues, the main consequence of having a non-uniform distribution is that the Jacobian will generally not have zero singular values. Instead, the singular value corresponding to tangent right singular vectors $\boldsymbol{v} \in \mathcal{M}_0$ will have a magnitude that depends inversely on the local variance of the data along that direction. While we cannot simply count the space corresponding to zero singular values, we can still quantify the dimensionality of the latent manifold by analyzing the time-dependency of gaps (i.e. sharp jumps) in the spectrum:

$$\Delta_{\mathrm{GAP}}^{(k)}(\boldsymbol{x}^*, t) = \lambda_{k+1}(\boldsymbol{x}^*, t) - \lambda_k(\boldsymbol{x}^*, t)\ . \tag{7}$$

In this case, the spectra of $J(\boldsymbol{x}^*, t)$ contain more information than just the local dimensionality of the manifold $\mathcal{M}_t$. Vectors in the tangent space $T_{\mathcal{M}}(\boldsymbol{x}^*, t)$ define locally linear sub-spaces with different variance. The variances of these local linear sub-spaces characterize the variability of the data along the corresponding directions.

## 5 Theoretical analysis

Here, we will provide a theoretical explanation of geometric memorization (dimensionality loss) using concepts from random-matrix theory and the statistical physics of disordered systems. To study the dynamics of the latent manifold evolution analytically, we will consider

data distributed according to the linear model $\boldsymbol{x}_0 = F\mathbf{z}$ where $\mathbf{z}$ is a $m$-dimensional standard Gaussian vector and $F$ is a $d \times m$ projection matrix. Equivalently, $\mathbf{x}_0 \sim \mathcal{N}(0, FF^\top)$. The choice of the linear model simplifies the statistical analysis while qualitatively capturing important features of the local phenomenology of the tangent spaces. In fact, at time $t$ the curvature of $\mathcal{M}_t$ is suppressed by the smoothing induced by the forward process, which, roughly speaking, linearizes the geometry at the same scale as our local Jacobian analysis (see Eq. (5)).

## 5.1 THE EXACT SCORE

In the linear Gaussian case, the Jacobian of the exact score function is independent of $\boldsymbol{x}$ and is given by the formula

$$J_t/t = \frac{1}{t}F\left[I_m + \frac{1}{t}F^\top F\right]^{-1}F^\top - I_d. \tag{8}$$

where we considered $J_t/t$ to control the divergence of the score at $t \to 0$. If the columns of $F$ are mutually orthogonal with $\|F_k\|_2^2 = \sigma^2$ for all indices, the singular spectrum of $J_t$ has a single gap at $d - m$, which encodes the dimensionality of the manifold. In a parallel paper (Anonymous, 2024), it is shown that the size of the gap at time $t$ is give by

$$\Delta_{GAP}(t;\sigma) = \frac{\sigma^2(1 + \alpha_m^{-1/2})^2}{t + \sigma^2(1 + \alpha_m^{-1/2})^2}. \tag{9}$$

where $\alpha_m = m/d$ denotes the ratio between the dimensionality of the manifold and the ambient dimensionality. While the gap is always present, in practice it only becomes distinguishable from the background noise at a finite time. Therefore, this formula indicates that subspaces with large variance emerge earlier during the generative reverse process. In the presence of $m$ different subspaces with dimensions $\{d_1, \ldots, d_m\}$ and variance $\{\sigma_2, \ldots, \sigma_m^2\}$, we will have one total manifold gap at $d - m$ and $m - 1$ intermediate gaps whose size approaches $(\sigma_k^{-2} - \sigma_{k+1}^{-2})/t$ for $t \to 0$. Only the total manifold gap remains for $t \to 0$ due to the $1/t$ normalization of the score. This reflects the fact that the component of the score orthogonal to the manifold diverges while the parallel component reaches a constant value, leaving us with: $\nabla_{\mathbf{x}} \log p_t(\mathbf{x}) \simeq \frac{1}{t}\left[\Pi - I_d\right]\mathbf{x}$. where $\Pi = F(F^\top F)^{-1}F^\top$ is the projector on the linear space $\mathcal{M}$.

## 5.2 THE EMPIRICAL SCORE

Computing the exact score function involves an average with respect to the true target distribution $p_0$. In real applications, we do not have direct access on $p_0$, whose behavior can only be inferred through a finite training set comprised of $N$ samples $\{\boldsymbol{y}^1, \ldots, \boldsymbol{y}^N\}$, with $\boldsymbol{y}^\mu \overset{iid}{\sim} p_0$. When training, we sample from the empirical distribution which we use as a proxy of the true distribution. The empirical distribution at time $t$ in the variance exploding framework is

$$p_t^N(\boldsymbol{x}) = \frac{1}{N\sqrt{(2\pi t)^d}}\sum_{\mu=1}^{N}e^{-\frac{\|\mathbf{x}-\mathbf{y}^\mu\|^2}{2t}} . \tag{10}$$

From the empirical distribution, we can write down the empirical score:

$$\nabla \log p_t^N(\boldsymbol{x}) = \sum_{\mu=1}^{N}w_\mu(\boldsymbol{x},t)\,(\boldsymbol{y}^\mu - \boldsymbol{x})\,/t\;, \tag{11}$$

where the weight $w_\mu(\boldsymbol{x},t) = p(\boldsymbol{y}^\mu \mid \boldsymbol{x})/\sum_{\nu=1}^{N}p(\boldsymbol{y}^\nu \mid \boldsymbol{x})$ is the posterior probability of the pattern $\boldsymbol{y}^\mu$ given the noisy state $\boldsymbol{x}$, were the possible states are restricted to the empirical set. This estimator is consistent, meaning that its bias approached the true score for $N \to \infty$. The random sampling of the dataset introduces statistical fluctuations that we can quantify by considering the estimator variance, which for large $N$ can be approximated as

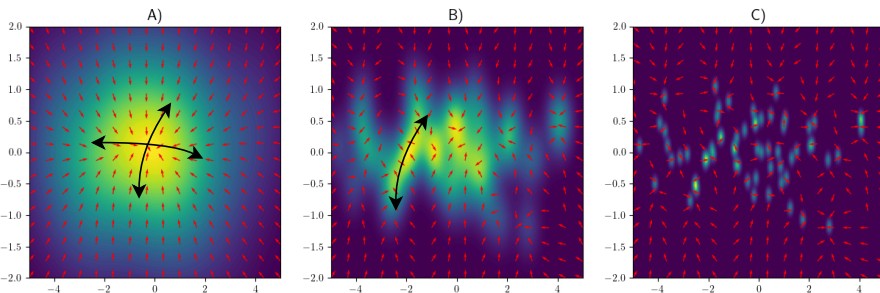

Figure 3: Visualization of the dimensionality loss phenomenon. Manifold sub-spaces with higher variance are lost due to 'condensation' (i.e. memorization). Panels A,B and C show the score estimated from a bivariate distribution with unequal variances for $\beta = 1$, $\beta = 10$ and $\beta = 100$ respectively. The red arrows show the empirical score while the heat-map visualizes the density.

$\text{var}\left[\nabla \log p_t^N(\boldsymbol{x})\right] \approx \text{var}(\boldsymbol{x}_0 \mid \boldsymbol{x})/\mathbb{E}\left[\tilde{N}_t(\boldsymbol{x})\right]$ , where $\text{var}(\boldsymbol{x}_0 \mid \boldsymbol{x})$ is the true posterior variance and $\tilde{N}_t(\boldsymbol{x}) = \left(\sum_{\mu=1}^N w_\mu^2(\boldsymbol{x}, t)\right)^{-1} \leq N$ is the effective number of data points used to estimate the score. When $t \to 0$, we always have that $\tilde{N}_t(\boldsymbol{x}) \to 1$, because the empirical score always fully memorizes in this limit. However, the empirical score exhibits generalization when the expected value is larger than the standard deviation induced by $\tilde{N}_t(\boldsymbol{x})$. Note that $\tilde{N}_t(\boldsymbol{x})$ is a function of the state $\boldsymbol{x}$ and that, consequently, the fluctuations in the empirical score depend on the "location" $\boldsymbol{x}$. This property is fundamental in our analysis of geometric memorization.

### 5.3 Memorization as glassy phase transition

The statistical behavior of the empirical score can be analyzed in the large $N$ limit by interpreting Eq. (10) as proportional to the partition function of a Random Energy Model (REM) (B. et al., 2024a; Lucibello and Mézard, 2024), which offers a simple model of disordered thermodynamic systems. The thermodynamic analysis of generative diffusion models is outlined in (Ambrogioni, 2023). In summary, each energy level $E_\mu$ is associated with a data point $\boldsymbol{y}^\mu$ in the training set and its energy depends on its Euclidean distance with the current state $\boldsymbol{x}_t$ (Ambrogioni, 2023), with the energy given by

$$E_\mu(\boldsymbol{x}) = -\frac{1}{2}\|\boldsymbol{y}^\mu\|^2 + \boldsymbol{x} \cdot \boldsymbol{y}^\mu \tag{12}$$

which leads to the partition function

$$Z_N(\boldsymbol{x}, t) = \sum_{\mu=1}^N e^{-\frac{1}{t}E_\mu(\boldsymbol{x})} \tag{13}$$

where the time parameter $t$ is analogous to the temperature of the system, which can be used to express the weights as a Boltzmann distribution: $w_\mu(\boldsymbol{x}, t) = \frac{1}{Z_N(\boldsymbol{x},t)}e^{-\frac{1}{t}E_\mu}$ . Since the empirical score is a Boltzmann average according to Eq. (13), studying its fluctuation under the random sampling of the data allows us to quantify the deviations from the exact score due to memorization effects. In our case, the energy levels are distributed according to

$$p(E; \boldsymbol{x}) = \int_{\mathbb{R}^n} \delta\left(E + \frac{1}{2}\|\boldsymbol{y}\|^2 - \boldsymbol{x} \cdot \boldsymbol{y}\right) dP_0(\boldsymbol{y}) \tag{14}$$

For small values of $t$ and large dataset sizes, the empirical score can be shown to be self-averaging, meaning that it is insensitive to the specific sampling of the training points, resulting to generalization of the underlying distribution. More formally, from the physical theory of REMs (Derrida, 1981), we know that, at the asymptotic limit of $d \to \infty$, the

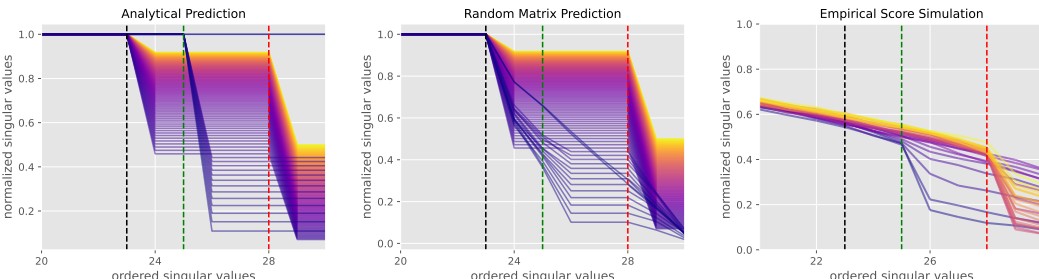

Figure 4: The ordered singular values of the Jacobian of the empirical score function of a linear manifold model as a function of the diffusion time $t$. Lighter colours are associated to larger times in the colour map. The parameters for the model are $d = 30$, $m = 7$, $\log(N)/d = 0.23$ with subspaces associated to variances $\sigma_1^2 = 1$ and $\sigma_2^2 = 0.3$ with dimensions $m_1 = 2$ and $m_2 = 5$ respectively. Left: approximated theoretical prediction in the memorization phase according to Eq. (19). Center: prediction from the approximated Jacobian in Eq. (18). Right: singular values obtained by the numerical measure of the Jacobian of the empirical score function (as described in Supp. B), evaluated from a synthetic data set of $N = 10^3$ points.

statistical system specified by Eq. (10) undergoes a random phase transition that separates a self-averaging high-temperature regime to a *condensation* regimes where Boltzmann averages depend on a small (i.e. sub-exponential) fraction of energy levels (Montanari and Mézard, 2009). In Supp. C, we show that, for $d$ much smaller that $N$, the condensation time for linear manifold data is, to a leading exponential order, equal to

$$t_c(\boldsymbol{x}) = \sqrt{\frac{d}{2\log(N)} \left( \frac{r_{4,\sigma}}{2} + \omega^2(\boldsymbol{x}) \right)} \, , \tag{15}$$

which demarcates the diffusion time when the empirical score becomes susceptible to fluctuations introduced by the random sampling of the dataset. In the formula, the term $r_{4,\sigma} = d^{-1} \sum_{i=1}^d \sigma_i^4$ captures the fluctuations in the norm of the data, while the directional quantity $\omega^2(\boldsymbol{x}) = d^{-1} \sum_{i=1}^d x_i^2 \sigma_i^2$ is the *variance density* along the direction $\boldsymbol{x}$. As we shall see, the balance between these two quantities plays a crucial role in geometric memorization effects. From standard REM calculations (see Supp. D), we can express the effective number of data points used to estimate the score at $\boldsymbol{x}$ at time $t$ as

$$\tilde{N}_t(\boldsymbol{x}) = \min\left( N, \frac{t}{1 - t_c^{-1}(\boldsymbol{x})} \right) \, . \tag{16}$$

where we introduced the minimum operator heuristically to account for the finite size of the system. The exact asymptotic theory is recovered for $N \to \infty$. Note that, since these quantities scale to the leading exponential order, they are therefore neglected quantities that scale sub-exponentially in $N$.

### 5.4 A theory of geometric memorization

The large $N$ analysis outlined in the previous sections give us a description of the fluctuations in the empirical score as a function of the state $\boldsymbol{x}$. The "spatial" dependency of these fluctuations ultimately depend on the data distribution $p_0(\boldsymbol{x})$, which outlines a rich geometric landscape that interacts in a complex way with the "spatial" variations in the exact score $\nabla \log p_t(\boldsymbol{x})$. To study the effect of this "spatially" non-homogeneous random fluctuations on the spectrum, we start from an approximate formula for the empirical score obtained by restricting the average to only $\tilde{N}_t(\boldsymbol{x})$ "active samples":

$$\nabla_{\boldsymbol{x}} \log p_t(\boldsymbol{x}) \approx \frac{1}{\tilde{N}_t(\boldsymbol{x})} \sum_{\mu=1}^{\tilde{N}_t(\boldsymbol{x})} (\boldsymbol{y}^\mu - \boldsymbol{x}) / t \tag{17}$$

where the "active samples" $\boldsymbol{y}^\mu$ are sampled from the posterior distribution $p(\boldsymbol{x}_0 \mid \boldsymbol{x}; t)$. In the linear Gaussian case, Eq. (17) follows a Normal distribution, since the posterior is itself

Normal. In the following, for the sake of simplicity, we will assume that $F$ is a diagonal $d \times d$ with diagonal entries $\sigma_k$, with $\sigma_k = 0$ for $d - m$ indices. This corresponds to a rotation to the basis of eigenvectors of $F^\top F$. If we assume that the fluctuations in the score are uncorrelated for a separation of the order of $\sqrt{t}$, we can quantify the statistical variability of the (smoothed) Jacobian (Eq. (5)) through the formula

$$J_{ij}(t) \sim \mathcal{N}\left(-\delta_{ij}\left(t + \sigma_i^2\right)^{-1}, \frac{\sigma_i^2}{t\left(t + \sigma_i^2\right)}\left[\phi(t, \mathbf{0}) + \phi(t, \boldsymbol{e}_j \cdot \sqrt{t})\right]\right) , \tag{18}$$

where we defined the function $\phi(t, \boldsymbol{x}) = \max\left(1/N, t^{-1} - t_c^{-1}(\boldsymbol{x})\right)$ . The expected value of this expression is just the Jacobian of the exact score, which determines the opening of the spectral gaps as explained in Sec. 5.1. On the other hand, in this model gaps can close due to the variance term. We can see this phenomenon qualitatively by considering the singular values spectrum of the expected value of Eq. (18):

$$\bar{s}_i = \sqrt{\frac{1}{\left(t + \sigma_i^2\right)^2} + \sum_{k=1}^{d} \frac{\sigma_k^2}{t^2\left(t + \sigma_k^2\right)^2}\left[\phi(t, \mathbf{0}) + \phi(t, \boldsymbol{e}_i \cdot \sqrt{t})\right]^2}. \tag{19}$$

Remember that we see a gap in the sorted spectrum if there is a large difference between two consecutive sorted singular values $s_k$ and $s_{k+1}$. This gap can disappear if I) $\phi(t, \boldsymbol{e}_k \cdot \sqrt{t})$ is larger than $\phi(t, \boldsymbol{e}_{k+1} \cdot \sqrt{t})$, or II) if the contribution of these variance terms make the contribution of the expected value negligible. Case I) is directional, as it depends on the direction of perturbations $\boldsymbol{e}_k$ and $\boldsymbol{e}_{k+1}$ and it leads to the selective suppression of a particular subspace. On the other hand, case II) is non-directional: it induces a synchronous suppression of all gaps and leads to complete memorization. The phenomenon of selective memorization is visualized in Fig. 3 for a two-dimensional distribution. For linear Gaussian, the closing times are determined by the critical time $t_c^{-1}(\boldsymbol{x})$, which itself depends on the constant term $r_{4,\sigma} = d^{-1}\sum_{i=1}^{d}\sigma_i^4$ and on the "directional" term $\omega^2(\boldsymbol{x}) = d^{-1}\sum_{i=1}^{d} x_i^2 \sigma_i^2$. This latter term is proportional to the variance along the subspace spanned by $\boldsymbol{x}$ and plays a crucial role in determining the differential disappearance of different subspaces at different times. Perhaps counter-intuitively, the subspace spanned by $\boldsymbol{e}_k$ is more vulnerable to memorization when $\omega^2(\boldsymbol{e}_k)$ is large. Therefore, subspaces that are more prominent in the distribution of the data and that emerge earlier during the diffusion process are also more vulnerable to memorization in the later stages of diffusion. This correspond to the form of feature memorization suggested in (Ross et al., 2024).

## 6 EXPERIMENTS

### 6.1 TESTING NUMERICALLY THE THEORY OF GEOMETRIC MEMORIZATION

Fig. 4 shows the evolution in time of the spectrum of the singular values obtained from Eq. (19) (left panel) Eq. (18) (central panel) and the direct computation of the empirical score introduced in section 5.2 (right panel). The experimental curves obtained from the empirical score look consistent with the theory, both in signaling the dimension of the subspaces and the opening times for the gaps, displaying the hierarchical structure associated to the ordering of the variances. Additional experiments are reported in Supp. D.1.

### 6.2 DIFFUSION NETWORKS TRAINED ON LINEAR MANIFOLD DATA

While our theory analyzes the empirical score, our experiments show that our results cast fundamental insight on geometric memorization in trained networks. Fig. 5 shows the spectra estimated from network trained on a linear manifold where the matrix $F$ is built such that datapoints live in two sub-spaces with variances equal to 1 (high variance) and 0.3 (low variance) respectively. The details of the experiment are given in Supp. E. We can compare these results with the spectra obtained in Fig. 4. The behavior of the trained network has several of the qualitative features predicted by our theoretical analysis. When $N$ is large, the spectra show the total manifold gap, as predicted by the exact theory. This gap remains present in the network even for $t = 1e^{-5}$, which shows that that trained network have a

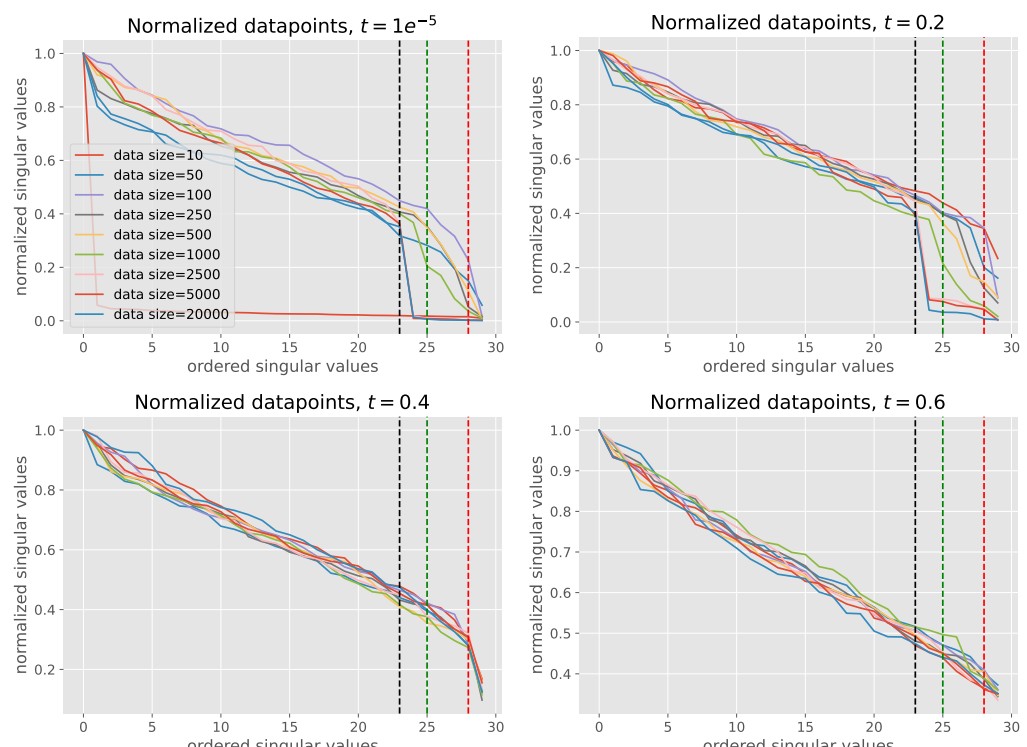

Figure 5: Spectra for different $t$ estimated from diffusion networks trained on linear Normal data with two subspaces with different variance using a range of dataset sizes. The red (green) dashed line correspond to the location of the theoretical spectral gap for the high (low) variance subspace. The black dashed line corresponds to the total manifold gap.

tendency to generalize, likely due to their finite capacity and by the implicit regularization induced by the parameterization. For intermediate values of $N$ (e.g. $N = 1000$), the theory predicts that, for small $t$, only the low variance gap should be observable, as the high variance sub-space is lost due to geometric memorization. Interestingly, this counterintuitive behavior seems to be present in the trained networks, where for $N = 250$ and $N = 500$, the drop in the spectrum is roughly aligned with the low-variance gap (green dashed line). Furthermore, the final generalization profile at $t = 1e^{-5}$ still behaves according to this prediction, suggesting that the ultimate generalization of the network can be predicted by the temporal dynamic of the empirical score. Finally, for small dataset sizes the behavior of the network disaligns from the theoretical prediction as the network returns to exhibit the high variance gap even for $t = 1e^{-5}$. This is not surprising since in this regime our large $N$ analysis is outside its domain of applicability, and the behavior of the network seems to reflect the global fit of a parameterized linear model. This is also visible in an additional experiment given in Supp. E.

### 6.3 GEOMETRIC MEMORIZATION IN NATURAL IMAGE DETASETS

We will now report the results of a series of experimental analysis where we trained on a series of increasingly large sub-datasets extracted from MNIST, Cifar10 and Celeb10. For each dataset size and time point, we estimate the latent dimensionality by locating the largest spectral gap and we study how dimensionality changes with the dataset size. The full details of dimensionality estimation and training are given in Supp. A and Supp. B. Fig. 6 shows the average spectra (left) and the average detected dimensionality (right) for the whole dataset (both train and test set), and see how that changes as a function of dataset size and diffusion time. Sharp spectral gaps are visible in the network trained on the MNIST dataset, where the estimated dimensionality increases sharply starting from 400 data points and reaches its peak at around 4000. The other datasets show less clear gaps in their spectra.

However, the location of their spectral inflection point decreases predictably as function of the dataset size, revealing an increasing trend in the estimated dimensionality. Interestingly, the dimensionality in Cifar10 does not seem to saturate, suggesting that the total dataset size still results in partial (geometric) memorization.

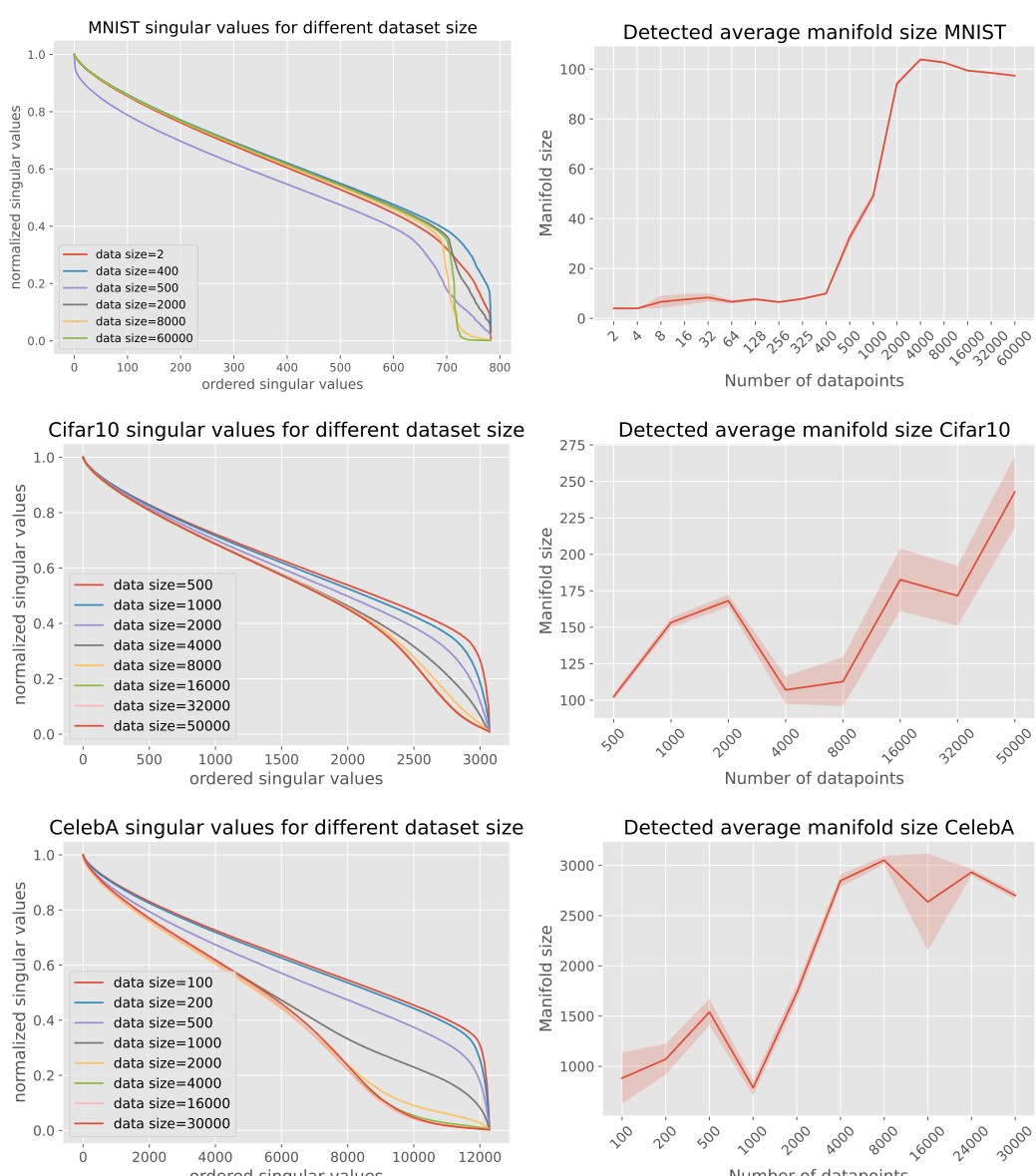

Figure 6: Spectra at $t = 1e^{-5}$ estimated on deep networks trained on natural image datasets with different dataset sizes. The estimated dimensionality tend to increase with the dataset size, suggesting a phenomenon of geometric memorization.

## 7 CONCLUSIONS

Our work opens the door for further analysis of generative diffusion using the tools of statistical physics, differential geometry and random matrix theory, which may cast light on generalization in these fascinating generative methods. Our theoretical analysis was focused on the empirical score function, further research may analyze theoretical Jacobian spectra in simple trained models and elucidate their deviations from the empirical theory.

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

## A EXPERIMENTAL METHODOLOGY: TRAINING AND MODEL ARCHITECTURE DETAILS

| Dataset | Image Size | Latent Dim. | Channel Mult. | Param. Count | Batch size | Iterations |
|---------|------------|-------------|---------------|--------------|------------|------------|
| Cifar10 | 32 | 128 | (1, 2, 2, 2) | 35.7M | 128 | 500,000 |
| Mnist | 28 | 128 | (1, 2, 2) | 24.5M | 128 | 400,000 |
| CelebA-HQ | 64 | 64 | (1, 1, 2, 2, 4, 4) | 27.4M | 64 | 800,000 |

Table 1: Table displaying both model and training configurations for each dataset.

For the toy examples, we train a Variance Exploding continuous score model with 2M training steps with batch size 128. We use a Residual Multi Layer Perceptron with hidden size of 128, with two residual blocks. Each block is composed by two linear layers with SiLu activation.

For the image models, we follow the diffusion setting in (Ho et al., 2020). We kept the variance scheduler, where $\beta_{\min} = 10^{-4}$ and $\beta_{\min} = 2 \times 10^{-2}$, the time steps $T = 1000$, and the score model backbone (PixelCNN++ (Salimans et al., 2017)) the same. In addition, for each of the datasets, we varied the model's channel multipliers, latent dimension, batch size, and training iterations to account for the complexity of the dataset and our available computing resources; see Table 1. For context, we primarily utilized NVIDIA Tesla V100 GPUs with 32 GB of memory for the training of our models.

For consideration of the data sizes we chosen for our experiment, we closely followed the setup of (Y. et al., 2023). Specifically, we first trained multiple diffusion models using different data split sizes from $\{0.5k, 1k, 2k, 4k, \ldots, |S|\}$, where $|S|$ is the full size of a given dataset. We trained our models without random flipping and utilized the exponential moving average version of the trained models, where we set the decay value to 0.9999 during training. For Cifar10 (Krizhevsky et al., 2014) and Mnist (Deng, 2012), we did not center-crop or resize the images. However, for CelebA-HQ (Liu et al., 2015), we center-cropped and downsized the images to $64 \times 64$ resolution. Moreover, we only use dropout for Cifar10 with the value of 0.1. To get a more comprehensive view of the reduction in the manifold size, in Fig. 6, we provide additional points for the low data size region. Specifically, for Mnist, we have $\{2, 4, 8, 16, 32, 64, 128, 256, 325, 400, 500\}$, and $\{100, 200\}$ for CelebA-HQ.

## B EXPERIMENTAL METHODOLOGY: MEASURING THE INTRINSIC DIMENSION OF THE DATA MANIFOLD

The method used to **geometrically visualize** the intrinsic dimension of the data manifold is the same as in (Stanczuk et al., 2022): the score function is measured across several independent positions in the vicinity of the manifold and ordered as the columns of a rectangular matrix $S$; the singular values of the matrix $S$ are computed and collected; the intrinsic dimension of the manifold is given by the $d - \ker(S)$, with the kernel is estimated directly from the spectrum of the singular values of $S$.

On the other hand, we propose a new method to **geometrically estimate** the intrinsic dimension of the data manifold. The procedure is based on empirically computing the absolute value of the second derivative of the singular values, selecting the first bigger value with respect to the median multiplied by a threshold factor. We further discard the initial singular value as it tends to be large, resulting in instabilities. We found this method to be more robust than the one proposed in (Stanczuk et al., 2022), especially for high dimensional datasets where there is no sharp drop in the spectrum of the singular values.

### B.1 COMPUTING THE SINGULAR VALUES OF THE JACOBIAN OF THE SCORE

For computing the singular values, we use the procedure described in (Stanczuk et al., 2022) reported in algorithm 1. For the linear models and MNIST models we used a symmetrized version which we empirically found to be more stable, reported in algorithm 2.

---

**Algorithm 1** Estimate singular values at $x_0$

---

**Require:** $s_\theta$ - trained diffusion model (score), $t_0$ - sampling time, $K$ - number of score vectors.
1: Sample $x_0 \sim p_0(x)$ from the data set
2: $d \leftarrow \dim(x_0)$
3: $S \leftarrow$ empty matrix
4: **for** $i = 1, ..., K$ **do**
5:     Sample $x_{t_0}^{(i)} \sim \mathcal{N}(x_{t_0}|x_0, \sigma_{t_0}^2 I)$
6:     Append $s_\theta(x_{t_0}^{(i)}, t_0)$ as a new column to $S$
7: **end for**
8: $(s_i)_{i=1}^d, (v_i)_{i=1}^d, (w_i)_{i=1}^d \leftarrow \text{SVD}(S)$

---

**Algorithm 2** Estimate singular values at $x_0$ with central difference

---

**Require:** $s_\theta$ - trained diffusion model (score), $t_0$ - sampling time, $K$ - number of score vectors.
1: Sample $x_0 \sim p_0(x)$ from the data set
2: $d \leftarrow \dim(x_0)$
3: $S \leftarrow$ empty matrix
4: **for** $i = 1, ..., K$ **do**
5:     Sample $x_{t_0}^{+(i)} \sim \mathcal{N}(x_{t_0}|x_0, \sigma_{t_0}^2 I)$
6:     Sample $x_{t_0}^{-(i)} \sim \mathcal{N}(x_{t_0}|x_0, -\sigma_{t_0}^2 I)$
7:     Append $\frac{s_\theta(x_{t_0}^{+(i)}, t_0) - s_\theta(x_{t_0}^{-(i)}, t_0)}{2}$ as a new column to $S$
8: **end for**
9: $(s_i)_{i=1}^d, (v_i)_{i=1}^d, (w_i)_{i=1}^d \leftarrow \text{SVD}(S)$

---

### B.2 COMPUTING THE INTRINSIC DIMENSION AT $x_0$

We report in 3 the algorithm used to find the intrinsic manifold dimension given the singular values. For MNIST we use $\bar{d} = 100$, $c = 15$ and $t = 0$, for Cifra10 $\bar{d} = 1000$, $c = 10$ and $t = 15$, and for CelebA $\bar{d} = 1000$, $c = 10$ and $t = 0$. Here $t$ correspond to the diffusion index in DDPM. We report an example of second derivative in Fig. 7.

---

**Algorithm 3** Estimate intrinsic manifold dimension at $x_0$

---

**Require:** $(s_i)_{i=1}^d$ from algorithms 1 or 2 - diffusion time, $t$ - datapoint size, $d$ - threshold, $c$ - discard values, $\bar{d}$.
1: $d_{svd}^2 \leftarrow |\frac{d^2}{ds} s_t[\bar{d} :] |$
2: $m \leftarrow \text{median}(d_{svd}^2)$
3: $n \leftarrow \arg \text{where}(d_{svd}^2 > c * m)$
4: $k \leftarrow d - n + \bar{d}$
5: **return** manifold dimension $k$

---

## C CONDENSATION TIME FOR POSITIONAL REM

For simplicity, we will perform the analysis for coordinate-aligned linear manifolds. Consider $d$-dimensional normally distributed vector-valued data $\boldsymbol{y}^\mu$ where each component $y_k^\mu$ follows a centered normal distribution with variance $\sigma_k^2$. In the linear manifold case, number $d - m$ of these variances is equal to zero, meaning that the distribution spans a $m$-dimensional linear manifold. The number of datapoints are taken to be exponential in the size of the ambient space, i.e. $N = \exp(\alpha d)$, with $\alpha > 0$. Notice that $\sigma_k^2$ correspond to the eigenvalues of $F^\top F$ in the random projection model and we assume we have changed the coordinate system. Let us take a fixed $\boldsymbol{x}$. Hence, in the variance exploding framework we have

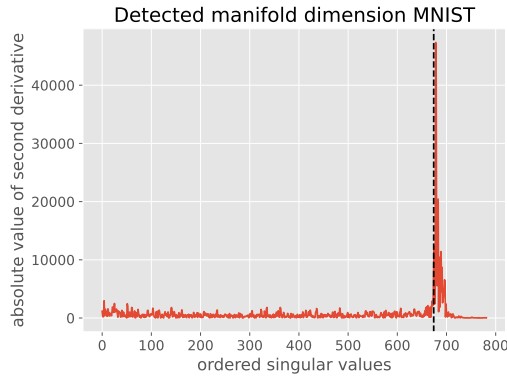

Figure 7: The figure shows the absolute values of the second derivatives computed with algorithm 3. The vertical line is the detected manifold dimension.

$$p_t(\boldsymbol{x}) = \frac{1}{N\sqrt{2\pi t}^d} \sum_{\mu=0}^{N} e^{-\frac{1}{2t}\|\boldsymbol{x}-\boldsymbol{y}^\mu\|^2} \tag{20}$$

$$= \frac{1}{N\sqrt{2\pi t}^d} e^{-\frac{\|\boldsymbol{x}\|^2}{2t}} \sum_{\mu=0}^{N} \exp\left(-\frac{1}{2t}\|\boldsymbol{y}^\mu\|^2 + \frac{1}{t}\boldsymbol{x}\boldsymbol{y}^\mu\right). \tag{21}$$

It is useful, at this point, to introduce the Random Energy Model (REM), firstly proposed by physicists (Derrida, 1981; Montanari and Mézard, 2009), now imported to computer science to characterize diffusion models (B. et al., 2024a). The REM consists in a collection of energy levels $\{E_\mu\}_{\mu \leq N}$ that interact with an external heat-bath at an inverse temperature $\beta$. The energy levels are random variables generated from a probability density function $p(E|\boldsymbol{\theta})$ where $\boldsymbol{\theta}$ can be some parameters of the model and source of disorder for the system. The thermodynamics of the model shows a condensation phase at a critical temperature $\beta_c$ that shares similarities with glassy transitions in spin-glass models (Mezard et al., 1986). Condensation, in turn, is analogous to memorization in diffusion models. The main thermodynamic quantities, such as the condensation temperature, can be fully recovered starting from the *partition function* of the system, given by

$$Z_N(\beta) = \sum_{\mu=1}^{N} e^{-\beta E_\mu}. \tag{22}$$

We can now map our diffusion model into a REM by redefining

$$\beta(t) = 1/t, \tag{23}$$

and

$$E_\mu(\boldsymbol{x}) = \frac{1}{2}\|\boldsymbol{x} - \boldsymbol{y}^\mu\|^2. \tag{24}$$

We call this model, *positional* REM, because the occurrence of condensation will depend on a position in the $d$-dimensional Euclidean space. Standard REM calculations are now performed to compute the free energy of the model and then the condensation time. The moment generating function of the energies is

$$\zeta(\lambda) = \lim_{d \to \infty} \frac{1}{d} \log \mathbb{E}_{\boldsymbol{y}} e^{-\frac{\lambda}{2t}\|\boldsymbol{y}\|^2 + \frac{\lambda}{t}\boldsymbol{x}\boldsymbol{y}} \tag{25}$$

$$= \lim_{d \to \infty} \frac{1}{d} \sum_{i=1}^{d} \log \int \frac{dy_i}{\sqrt{2\pi\sigma_i^2}} \exp - \frac{y_i^2}{2}\left(\frac{1}{\sigma_i^2} + \frac{\lambda}{t}\right) + \frac{\lambda}{t}x_i y_i \tag{26}$$

$$= \lim_{d \to \infty} \frac{1}{d} \left[ -\frac{1}{2}\sum_{i=1}^{d} \log\left(1 + \lambda\frac{\sigma_i^2}{t}\right) + \frac{\lambda^2}{2t^2}\sum_{i=1}^{d} \frac{x_i^2 \sigma_i^2}{1 + \lambda\frac{\sigma_i^2}{t}} \right] \tag{27}$$

The derivative of the zeta function is

$$\zeta'(\lambda) = \lim_{d \to \infty} \frac{1}{d} \left[ -\frac{1}{2t}\sum_i \frac{\sigma_i^2}{1 + \lambda\frac{\sigma_i^2}{t}} + \frac{\lambda}{t^2}\sum_i \frac{x_i^2 \sigma_i^2}{1 + \lambda\frac{\sigma_i^2}{t}} - \frac{\lambda^2}{2t^3}\sum_i \frac{x_i^2 \sigma_i^4}{(1 + \lambda\frac{\sigma_i^2}{t})^2} \right]. \tag{28}$$

At large times, $\zeta(\lambda)$ and $\zeta'(\lambda)$ become respectively

$$\zeta(\lambda) = -\frac{\lambda}{2t}r_{2,\sigma} + \frac{\lambda^2}{4t^2}r_{4,\sigma} + \frac{\lambda^2}{2t^2}\omega^2(\boldsymbol{x}), \tag{29}$$

$$\zeta'(\lambda) = -\frac{\lambda}{2t}r_{2,\sigma} + \frac{\lambda^2}{2t^2}r_{4,\sigma} + \frac{\lambda^2}{t^2}\omega^2(\boldsymbol{x}). \tag{30}$$

Where

$$r_{2,\sigma} = \lim_{d \to \infty} \frac{1}{d}\sum_i \sigma_i^2 \tag{31}$$

$$r_{4,\sigma} = \lim_{d \to \infty} \frac{1}{d}\sum_i (\sigma_i^2)^2 \tag{32}$$

$$\omega^2(\boldsymbol{x}) = \lim_{d \to \infty} \frac{1}{d}\sum_i (x_i)^2 \sigma_i^2. \tag{33}$$

The condition for the condensation time is $\alpha + \zeta(1) - \zeta'(1) = 0$, from which we obtain

$$t_c(\boldsymbol{x}) = \sqrt{\frac{\frac{r_{4,\sigma}}{2} + \omega^2(\boldsymbol{x})}{2\alpha}}. \tag{34}$$

As clear from the formula, this time depends on the variance $\omega^2(\boldsymbol{x})$ along the direction of $\boldsymbol{x}$. This implies that, when $\boldsymbol{x}$ is aligned to a linear sub-manifold with higher variance, condensation around this state will happen earlier, leading to a decrease in the estimated commonality of the latent manifold. Fig. 8 shows a comparison between the exact approach for computing $t_c(x)$ (i.e. using Eqs. (25), (28)) and the small $\alpha$ expansion (i.e. Eq. (15)), showing a good qualitative agreement between the two quantities at all values of $\alpha$. The right panel of the same figure also displays a strong dependence of the exactly computed condensation time.

If each dimension has equal variance $\sigma^2$, the directional variance density is just $\sigma^2$, which implies that the critical condensation time depends linearly on the dimensionality but only logarithmically on the number of data points. This implies that in isotropic case, in order to avoid condensation an exponential number of data points is needed. However, if only $\alpha_m$ dimensions have non-zero variance, it is straightforward to see that the exponential dependency will scale with $\alpha_m$ instead of $m$. More generally, the exponential scaling depends on the total variance $m \omega(\boldsymbol{x})$, which implies that it is realistic to learn high-dimensional spaces as far as most of these dimensions have vanishing variance.

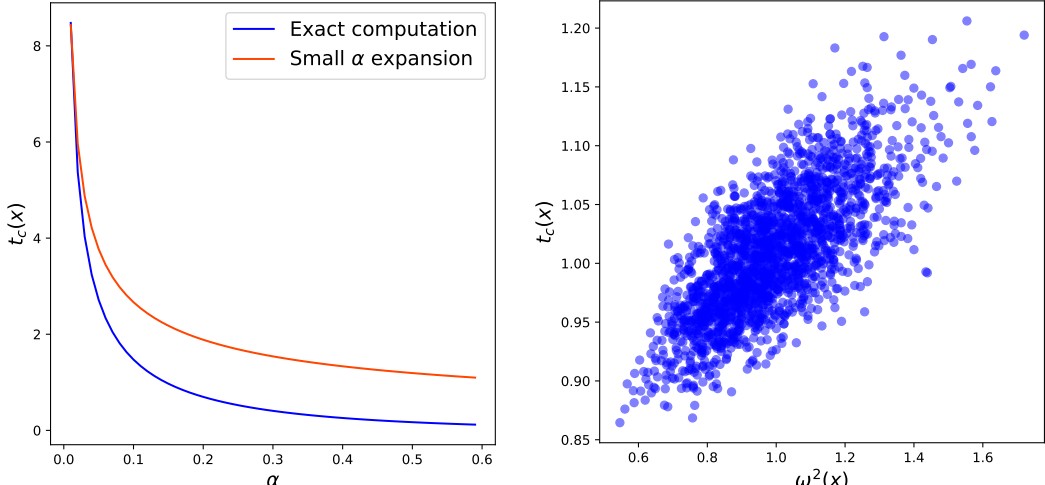

Figure 8: Condensation time as a function of position $x$ computed according to the REM calculations. Left: we have generated one single position $\boldsymbol{x}$ in a ambient space of dimension $d = 100$ and one single matrix $F$ of dimensions $100 \times 50$ (with $m = 50$ dimension of the latent space). Both $\boldsymbol{x}$ and $F$ are generated according to a Gaussian process with zero mean and unitary variance; we show the comparison between the exact calculation of the positional condensation time and the approximated version that is fully explicit in the directional variance $\omega^2(\boldsymbol{x})$. Right: we generate 2000 random positions $\boldsymbol{x}$ around the origin of the ambient space of dimension $d = 100$; the latent space dimension is $m = 50$ and $\alpha = 0.15$; we show the dependence of the exact positional condensation time as a function of $\omega^2(\boldsymbol{x})$, showing a qualitatively similar behaviour with respect to the approximated expression of $t_c$.

## D Analysis of the empirical Jacobian

We can relate this random energy analysis to the spectra of Jacobian eigenvalues using a heuristic argument. In the linear manifold example, the Jacobian of the true score function at $t = 0$ is diagonal with eigenvalues equal to $-1/\sigma_k^2$. This results in spectral gaps when different sub-spaces have different variances. For a finite value of the inverse temperature $\beta(t) = 1/t$, the eigenvalues are $-1/\sigma_k^2 - \beta$. After the critical condensation time, the empirical score gives a good approximation of the true score. On the other hand, in the condensation phase the empirical score is dominated by the (quenched) fluctuations in the data distribution. First, we can introduce the participation ratio

$$Y(\beta, \boldsymbol{x}) = \frac{Z(2\beta, \boldsymbol{x})}{Z(\beta, \boldsymbol{x})^2} \ . \tag{35}$$

This thermodynamic quantity can be roughly interpreted as the inverse of the number of energy levels with non-vanishing weights. In the condensation phase, this will be a finite number while it becomes infinite in the high temperature phase.

In the thermodynamic limit and for $\beta(t) \geq \beta_c(t)$, the participation ratio of our REM model is given by

$$\mathbb{E}[Y(\beta, \boldsymbol{x})] = 1 - \frac{\beta_c(t, \boldsymbol{x})}{\beta(t)} \ , \tag{36}$$

which implies that the number of datapoints that contribute to the score function at $\boldsymbol{x}$ is

$$\tilde{N} = e^{\alpha \tilde{d}(\beta, \boldsymbol{x})} = 1/Y(\beta, \boldsymbol{x}) = \frac{\beta(t)}{\beta(t) - \beta_c(t, \boldsymbol{x})} \ . \tag{37}$$

Note that this number tends to one for $\beta(t) \to \infty$, meaning that in the low-time limit the score depends on a single pattern.

In this phase, the score is dominated by approximately $e^{\alpha \tilde{d}(\beta, \boldsymbol{x})} = \frac{\beta(t)}{\beta(t) - \beta_c(t, \boldsymbol{x})}$, leading to the expression

$$\nabla_{\boldsymbol{x}} \log p_t(\boldsymbol{x}) \approx \frac{\beta}{e^{\alpha \tilde{d}(\beta, \boldsymbol{x})}} \sum_{\mu=1}^{e^{\alpha \tilde{d}(\beta, \boldsymbol{x})}} (\boldsymbol{y}^\mu - \boldsymbol{x}) \tag{38}$$

where $\boldsymbol{y}^\mu \sim p(\boldsymbol{y}^\mu \mid \boldsymbol{x}, \beta) \propto e^{-\boldsymbol{y}^T(\Lambda^{-1} + \beta I_d)\boldsymbol{y}/2 + \beta \boldsymbol{x} \cdot \boldsymbol{y}}$. Therefore, the empirical score approximately follows the distribution

$$\nabla_{\boldsymbol{x}} \log p_t(\boldsymbol{x}) \sim \mathcal{N}\left(-M(\beta)\boldsymbol{x}, \beta(\Lambda^{-1} + \beta I_d)^{-1} \max(0, \beta - \beta_c(\boldsymbol{x}))\right) . \tag{39}$$

where $M(\beta) = \beta(\Lambda^{-1} + \beta I_d)^{-1}\Lambda^{-1}$ and $\Lambda$ being the diagonal matrix collecting the variances $\sigma_k^2$ and we used the fact that $e^{\alpha \tilde{d}(\beta, \boldsymbol{x})} = \beta/(\beta - \beta_c(\boldsymbol{x}))$. The minimum in the formula is due to the fact that, for $\beta < \beta_c$, an exponentially large number of patterns participate in the estimation of the score, which leads to a complete suppression of the variance. On the other hand, the variance of the empirical score estimator diverges for $\beta \to \infty$. In fact, during condensation, the fluctuations in the random sampling of the datapoints are not suppressed due to the small number of non-vanishing weights.

We can finally estimate the distribution of the eigenvalues estimated from the empirical Jacobian matrix. Let us set ourselves on $\boldsymbol{x} = \boldsymbol{0}$ and perturb along the directions of the eigenvectors of $F^\top F$. We estimate the elements of the Jacobian of the score function with respect to the orthogonal direction $\boldsymbol{e}_j$ using a perturbative approach, i.e.

$$J_{ij}(\beta) \approx \sqrt{\beta}\left(\partial_{x_i} \log p_t(\boldsymbol{e}_j/\sqrt{\beta}) - \partial_{x_i} \log p_t(\boldsymbol{0})\right). \tag{40}$$

Using Eq. (39), we can then write an approximate distribution for the elements of the Jacobian as

$$J_{ij}(\beta) \sim \mathcal{N}\left(-\beta \delta_{ij}\left(1 + \beta \sigma_i^2\right)^{-1}, \beta^2\left(\sigma_i^{-2} + \beta\right)^{-1}\left[\phi(\beta, \boldsymbol{0}) + \phi(\beta, \boldsymbol{e}_j/\sqrt{\beta})\right]\right) . \tag{41}$$

where we assumed that the fluctuations in $\nabla_{\boldsymbol{x}} \log p_t(\boldsymbol{e}_j/\sqrt{\beta})$ are independent from the fluctuations in $\nabla_{\boldsymbol{x}} \log p_t(\boldsymbol{0})$ and $\nabla_{\boldsymbol{x}} \log p_t(\boldsymbol{e}_k/\sqrt{\beta})$ for all $k$s. In this expression, we introduced the function

$$\phi(\beta, \boldsymbol{x}) = \max\left(0, \beta - \beta_c(\boldsymbol{x})\right) . \tag{42}$$

We can now recover the singular values of $J(\beta)$ as minus the square roots of the eigenvalues of $J(\beta)^\top J(\beta)$. In general, the matrix $J(\beta)^\top J(\beta)$ can have a complex spectral distribution. An approximate formula for the singular values of $J(\beta)$ is

$$s_i \approx -\sqrt{\beta^2(1 + \beta \sigma_i^2)^{-2} + \beta^4 \sum_{k=1}^d \left(\sigma_k^{-2} + \beta\right)^{-2}\left[\phi(\beta, \boldsymbol{0}) + \phi(\beta, \boldsymbol{e}_i/\sqrt{\beta})\right]^2}. \tag{43}$$

To obtain this formula, we write $J$ as

$$J = A + B \tag{44}$$

where $A$ is a diagonal matrix corresponding to the mean of Eq. (41), while $B$ corresponds to the variance. Therefore, $J^\top J$ becomes

$$J^\top J = A^\top A + A^\top B + B^\top A + B^\top B. \tag{45}$$

This expression is dominated by the two symmetric terms, so we can write

$$J^\top J \approx A^\top A + B^\top B. \tag{46}$$

Then, the term $A^\top A = A^2$ is, of course, still diagonal, while the term $B^\top B$ is diagonally dominant. Calling $C = \sum_{ik} B_{ik} B_{ik}$, we can approximate the singular values as $\sqrt{A^2 + C^2}$, obtaining Eq. (43). Note however that the distribution of the spectrum does not concentrate exactly to Eq. 43 in the large $N$. Nevertheless, Eq. 43 gives an accurate picture of the qualitative behavior, as shown in Sec. D.1.

These results also show that in some regimes Eq. 43 is more in agreement with the numerical empirical score than the correct spectrum of Eq. 41, which is likely due to the fact that Eq. 41 overestimates the fluctuations by ignoring the correlations of the score at different points.

## D.1 NUMERICAL ANALYSIS

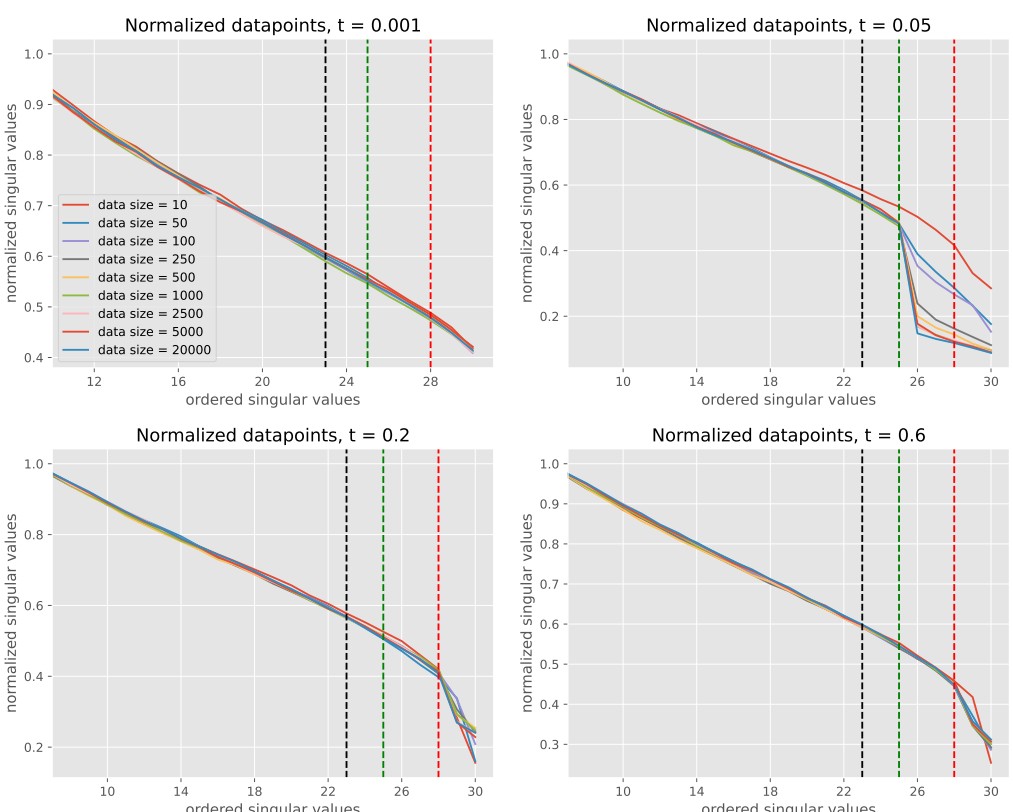

Figure 9: Ordered singular values of the Jacobian of the empirical score in the case of the linear data model. The parameters for the model are $d = 30$, $m = 7$, $\alpha = \log(N)/d = 0.23$ with a subspace associated to a variance $\sigma_1^2 = 1$ of dimension $m_1 = 2$ and another subspace with variance $\sigma_2^2 = 0.3$ and dimension $m_2 = 5$. Different lines are associated to different sizes of the training set. Measures have been averaged over 30 realizations of the experiment.

In order to test our theory of the empirical score, we plot the singular values of the Jacobian in an ordered fashion, as done experimentally in the previous sections: this allows to visualize the drops forming due to the described memorization phase transition. As a first test, a set of $N$ data have been generated according to the linear manifold model with two variances, and the empirical score has been computed out of these points as in Eq. (10). Therefore, we measured the Jacobian according to same method used for the trained models, described in section B. The condensation time appearing in the formulas has been computed according to the method explained in Supp. C. Figs. 9 and 10 report the profiles of the ordered spectra at different times when data-sets of different sizes are employed to assemble the empirical score. We notice the same phenomenology of the gaps predicted by the theory: the gaps indicating the dimensions of both the two subspaces progressively open starting from the largest variance and ending to the smallest one.

As a second experiment we confront the evolution of the ordered singular values in time, as obtained from three methods: the functional form in Eq. (19); by extraction from the random matrix expressed in Eq. (18); by computing the empirical score function from a synthetic set of $N$ datapoints (i.e. as performed in the previous experiment). The time evolution of the gaps is reported in figures 11 and 12 for two choices of the parameters of the model. We conclude that both the random matrix prediction and the simulation capture the phenomenology predicted by the analytical expression of the singular values obtained in Eq. (19).

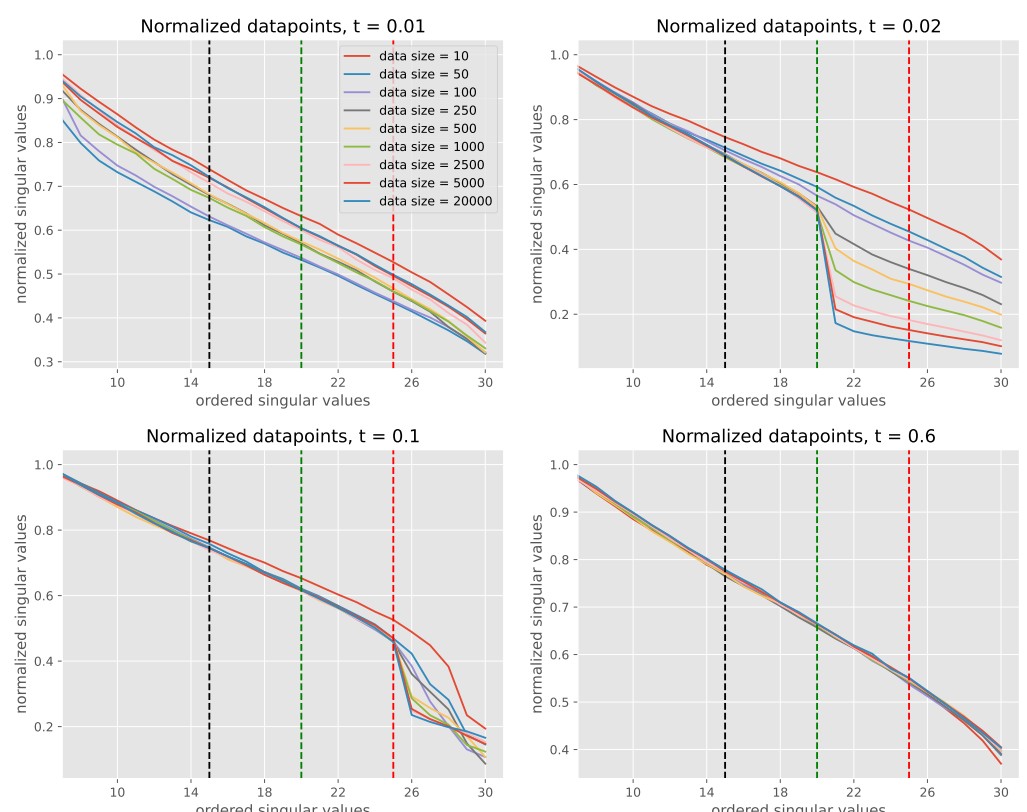

Figure 10: Ordered singular values of the Jacobian of the empirical score in the case of the linear data model. $d = 30$, $m = 15$, $\alpha = \log(N)/d = 0.23$ with a subspace associated to a variance $\sigma_1^2 = 1$ of dimension $m_1 = 5$ and another subspace with variance $\sigma_2^2 = 0.3$ and dimension $m_2 = 10$. Different lines are associated to different sizes of the training set. Measures have been averaged over 30 realizations of the experiment.

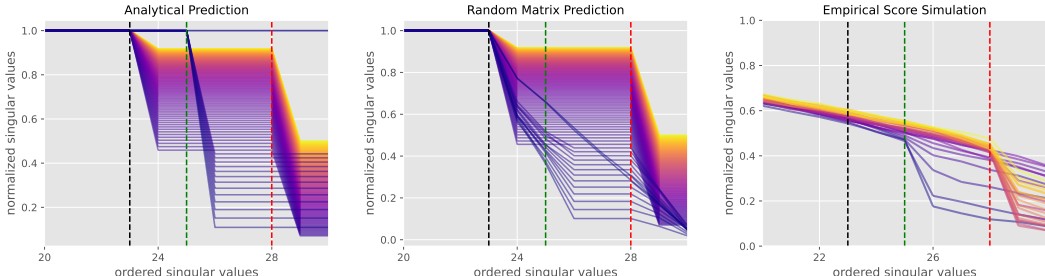

Figure 11: The ordered singular values of the Jacobian of the empirical score function of a linear manifold model as a function of the diffusion time $t$. Lighter colours are associated to larger times in the colour map. The parameters for the model are $d = 30$, $m = 7$, $\alpha = \log(N)/d = 0.23$ with a subspace associated to a variance $\sigma_1^2 = 1$ of dimension $m_1 = 2$ and another subspace with variance $\sigma_2^2 = 0.3$ and dimension $m_2 = 5$. Left: approximated theoretical prediction in the memorization phase according to Eq. (19). Center: prediction from the approximated Jacobian in Eq. (18). Right: singular values obtained by the numerical measure of the Jacobian of the empirical score function (as described in section B), evaluated from a synthetic data set of $N = 10^3$ points.

Finally, figures 13 and 14 report the evolution of the gaps according to, respectively, the closed formula for the singular values in Eq. (19) and the approximated random matrix in Eq. (10), when we change the size of the data-set.

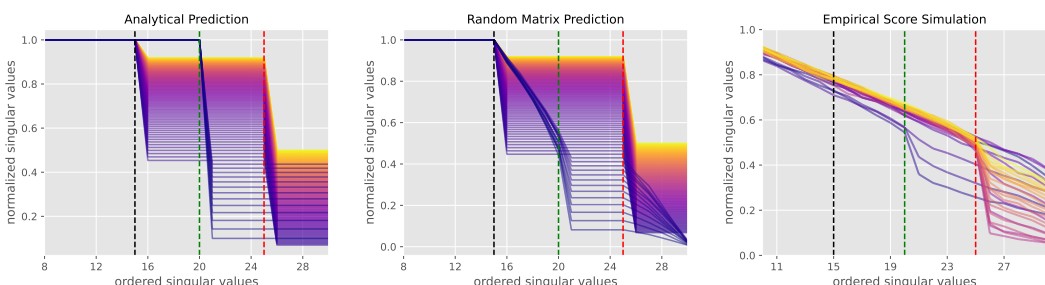

Figure 12: The ordered singular values of the Jacobian of the empirical score function of a linear manifold model as a function of the diffusion time $t$. Lighter colours are associated to larger times in the colour map. The parameters for the model are $d = 30$, $m = 15$, $\alpha = \log(N)/d = 0.23$ with a subspace associated to a variance $\sigma_1^2 = 1$ of dimension $m_1 = 5$ and another subspace with variance $\sigma_2^2 = 0.3$ and dimension $m_2 = 10$. Left: approximated theoretical prediction in the memorization phase according to Eq. (19). Center: prediction from the approximated Jacobian in Eq. (18). Right: singular values obtained by the numerical measure of the Jacobian of the empirical score function (as described in section B), evaluated from a synthetic data set of $N = 10^3$ points.

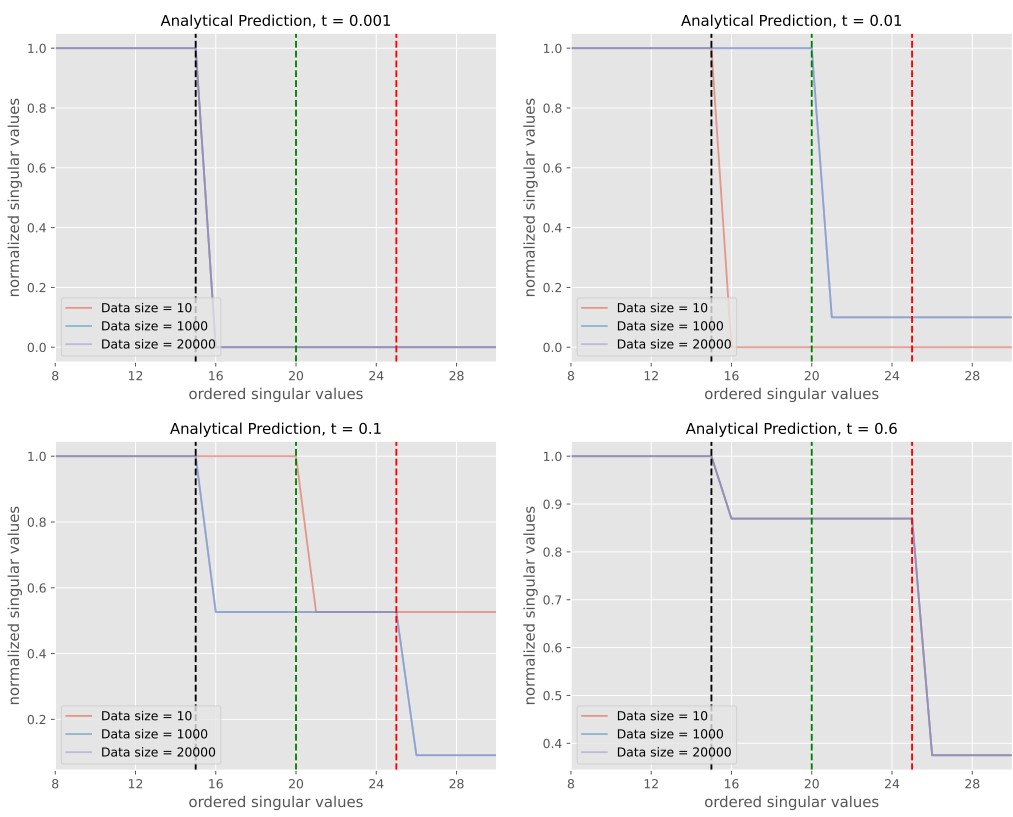

Figure 13: Ordered singular values of the Jacobian of the empirical score in the case of the linear data model estimated from Eq. (19). The parameters for the model are $d = 30$, $m = 7$, $\alpha = \log(N)/d = 0.23$ with a subspace associated to a variance $\sigma_1^2 = 1$ of dimension $m_1 = 2$ and another subspace with variance $\sigma_2^2 = 0.3$ and dimension $m_2 = 5$. Different lines are associated to different sizes of the training set.

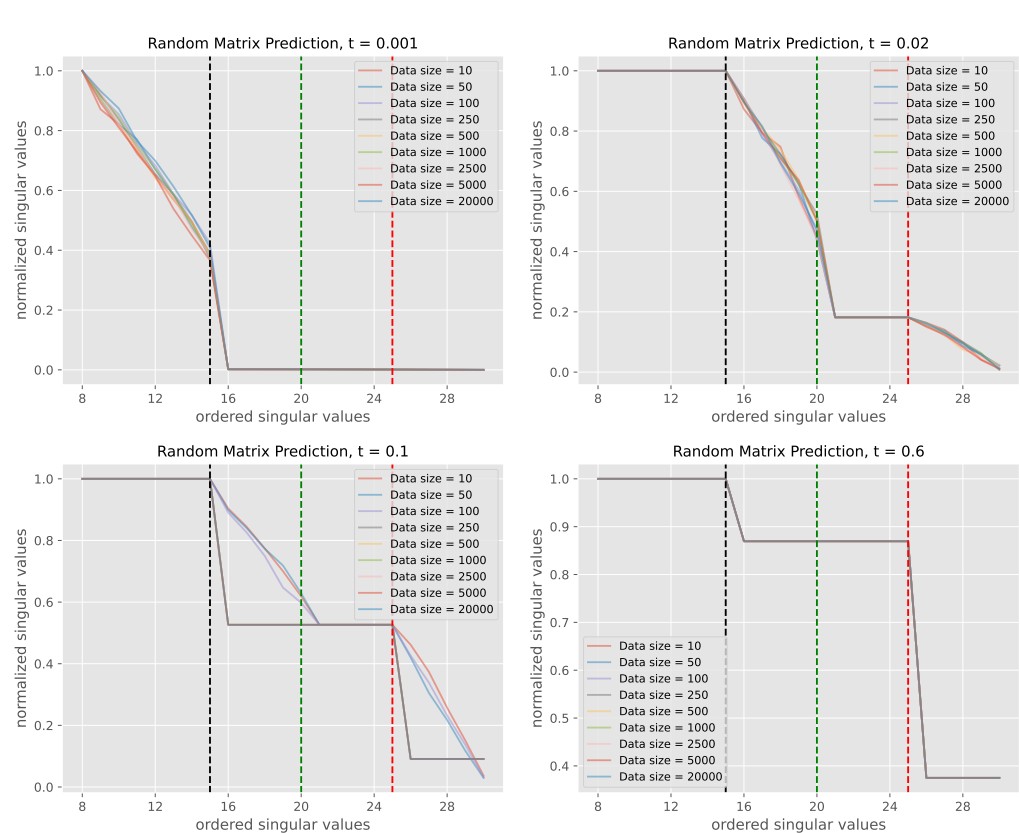

Figure 14: Ordered singular values of the Jacobian of the empirical score in the case of the linear data model, estimated from the random matrix in Eq. (10). $d = 30$, $m = 15$, $\alpha = \log(N)/d = 0.23$ with a subspace associated to a variance $\sigma_1^2 = 1$ of dimension $m_1 = 5$ and another subspace with variance $\sigma_2^2 = 0.3$ and dimension $m_2 = 10$. Different lines are associated to different sizes of the training set.

## E  Additional experimental results on trained networks

In addition to the experiments described in section 6 we report here some tests on synthetic data generated on the linear manifold introduced in section 5.

First, we report numerical results from experiments in an ambient space of dimension $d = 100$, while the manifold lives in a space of dimension $m = 40$. On the same manifold, through the choice of a diagonal $F$ matrix, we define subspaces of different variances, which will result in the opening of gaps at different times in the spectrum of the singular values of the matrix obtained by sampling the score functions. In the specific, we choose the case of two subspaces associated to two variances of the data and the particular scenario of $m$ different variances sampled uniformly at random. In Fig. 15 we plot the spectra at the smallest diffusion time $\epsilon$, for models trained on datasets with different amount of training samples. With few training samples, we cannot see any gap opening. However, as the training samples increase, the model starts generalizing to the sub-spaces with higher variance, indicating both a smooth transition between generalization and memorization, and that the subspaces with higher variance are learned first by the model. As we shall see, as predicted by the theory the network will instead generalize to subspaces of low variance for parameter settings where $\sigma_2^2$ is lower than $\sigma_1^2$ but not negligible. Furthermore, in Fig. 16 report a similar experiment with a smaller ambient dimension, i.e. $d = 30$. Now the geometric memorization phenomenon is more evident for medium data-set sizes at small times. Moreover, the phenomenology emerging from the trained model is fully consistent with the one resulting from the empirical score, obtained through the same choice of the parameters, as showed in figures 12, 13 and 14. This conclusion suggests two powerful insights about diffusion models: the network behaviour is consistent with our theory of memorization derived from the physics of Random Energy Models; the trained score function behaves consistently with the empirical score for a certain choice of the parameters.

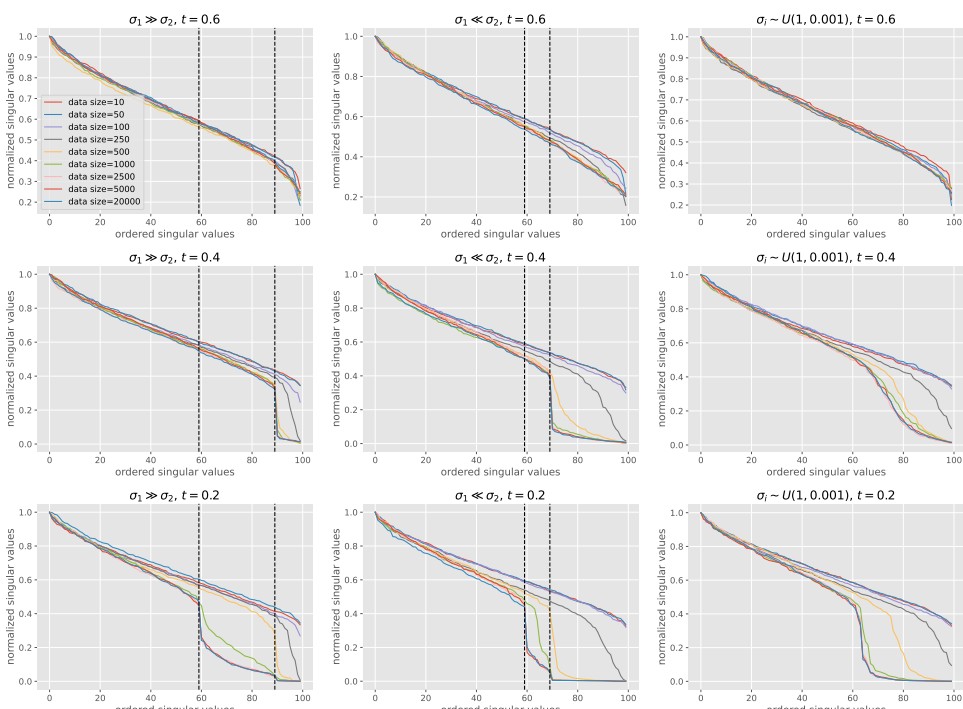

Figure 15: Singular values for models trained on different number of samples from the dataset (in the legend) at $t = 0.6$, $t = 0.4$ and $t = 0.2$, from top to bottom respectively. From left to right: model with $\sigma_1^2 = 1$, $\sigma_2^2 = 0.01$; model with $\sigma_1^2 = 0.01$, $\sigma_2^2 = 1$; model with uniformly sampled variances.

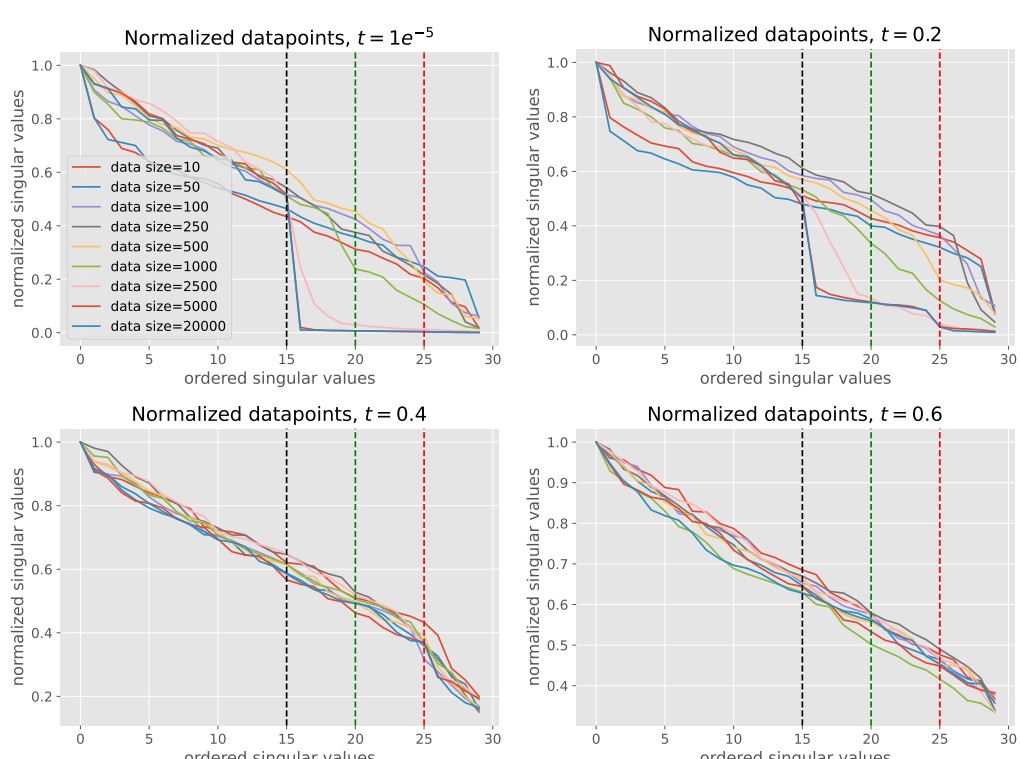

Figure 16: Ordered singular values of the Jacobian of the trained score function in the case of the linear data model. $d = 30$, $m = 15$, $\alpha = \log(N)/d = 0.23$ with a subspace associated to a variance $\sigma_i^2 = 1$ of dimension $m_1 = 5$ and another subspace with variance $\sigma_2^2 = 0.3$ and dimension $m_2 = 10$. Different lines are associated to different sizes of the training set.

