# OpenReview forum: "Losing dimensions: Geometric memorization in generative diffusion"
_ICLR.cc/2025/Conference — ICLR 2025 Conference Withdrawn Submission_

### Official Review · Reviewer_eZG7 · 2024-10-28

**Soundness:** 3
**Presentation:** 2
**Contribution:** 2
**Rating:** 5
**Confidence:** 4

**Summary:**

This paper analyzes the sampling dynamics in diffusion models by studying the Jacobian of the score function. The authors' main contribution is introducing a **spectral gap** between the leading singular values, which correspond to score components in the orthogonal space, and the trailing singular values, corresponding to the tangent space.

The study examines the spectral gap in two main settings. First, they consider the exact score, which is always defined by the linear manifold, making the spectral gap easy to analyze. They then extended this analysis to the empirical score which is a Boltzmann average, applying statistical physics to investigate the "active samples" that influence the score and the spectral gap. This approach allows the authors to characterize how the spectral gap (and its associated subspaces) responds to memorization effects.

The authors provide a rigorous analysis using synthetic linear Gaussian data and validate their findings with real-world datasets. They further propose to estimate the manifold dimension using such spectral gap.

**Strengths:**

1. The authors provide a rigorous and insightful analysis of the empirical score, observing that subspaces with larger variance exhibit a earlier condensation time $t_c$ and are more vulnerable to memorization.

2. They also clearly visualize how the spectral gap opens as the number of samples increases and how it evolves across different time steps (Figure 5), offering a new perspective on the phase transition from memorization to generalization.

**Weaknesses:**

1. The ideas in this paper—such as the spectral gap, critical time, "active" samples, generalization during sampling of memorized models, and geometric memorization—are too dense and interwoven, making the paper somewhat difficult to follow. Prioritizing key results could improve clarity.

2. The paper offers limited insight beyond the observed phenomena, as memorized diffusion models are impractical in real-world applications and have been extensively studied in prior work (such as https://arxiv.org/abs/2402.18491). This work appears largely as a technical extension of that research.

**Questions:**

1. Could you clarify the claim "empirical score exhibits generalization when the expected value is larger than the standard deviation induced by $\tilde{N}_t(X)$" (lines 292–293)? Is this the same as the conventional understanding of generalization, typically defined for generation results? Also, if the spectral gap opens at a particular time step $t$ (e.g. $t=1e^{-5}$), can we consider the diffusion model to be operating in the generalization regime?

2. The manifold dimension estimates shown in Figure 6 seem inconsistent with existing results (e.g., https://arxiv.org/abs/2104.08894). Specifically, a dimension of 100 for MNIST and 3000 for CelebA appears quite high.

---

### Official Review · Reviewer_3sh3 · 2024-11-02

**Soundness:** 3
**Presentation:** 2
**Contribution:** 4
**Rating:** 6
**Confidence:** 3

**Summary:**

Coming from the probabilistic side of generative modelling, learning about the associative memory perspective and this paper was very insightful! The paper presents a nice view of the memorization/generalization dichotomy in diffusion models and aims to formalize "geometric memorization" --- a scenario in which the model learns a lower-dimensional manifold (but not 0-dimensional) than the true data distribution ---  and validate whether or not it occurs in practice.

At a high level, I have two main reservations. First, and perhaps most importantly, the presentation and organization of the paper can be improved, which I will elaborate on below. Second, while the theoretical grounding is solid, the experiments focus mostly on MNIST/CIFAR10/CelebA. In my experience, memorization in these datasets often results from data duplication and exact memorization. More interesting cases arise with models like Stable Diffusion, where features of an image are memorized, while other generated aspects are novel (generalization). Including such experiments would certainly improve the paper; however, I am well aware of the computational limitations of SVD decompositions for high-dimensional data.

All in all, this is a technically solid paper and I will definitely recommend acceptance! However, the presentation can be greatly improved for better accessibility. In what follows, I have provided explicit examples of places where the paper can be improved. If most of these are resolved, I would happily raise my score!

**Strengths:**

1. The paper is theoretically strong, offering useful insights into generalization and memorization in diffusion models.

2. The visualizations, especially Figures 1 and 3, greatly aid understanding.

3. The connections to statistical physics are impressive and definitely merit broader communication within the ML community.

**Weaknesses:**

1. The experiments are limited. In fact, I am slightly concerned about how non-linear manifolds behave in practice. Could you please provide the same analysis conducted in Figure 5, but instead of using linear manifolds embed that in a high-dimensional sphere?

2.  The formal explanation of 5.1 can be greatly improved! Please explicitly define the higher and low-dimensional subspaces that are referred to in the rest of the text. In other words, please explicitly define what these subspaces are in terms of $F$. To my understanding, these subspaces correspond to the subspace obtained by eigenvectors of $FF^\top$ that have eigenvalues above different thresholds. Nonetheless, it is important to explicitly define it to avoid confusion.

3. For better accessibility for the ML community, this is a paper that can benefit a lot by having a section such as "a primer on random energy models". For example, being an ML researcher with little knowledge of statistical physics, I was unable to follow the exposition in Appendix C leading up to eq. (15) in the limited timeframe for review.

## Extra Points

1. Figure 1 is not referenced. It would be useful to incorporate it into the explanation, especially when the latent manifolds are defined.

2. Please add the explicit derivations of eq. (8) to the Appendix of the paper. What I am getting is different. Also, this equation does not match eq. (14) in the supplementary material.

3. Related to 5.1, this is probably a comment for the supplementary provided, but Kamkari et al. [A] and Tempczyk et al. [B] do an experiment which is somewhat different but gets the same conclusion. Please refer to the paragraph "FLIPD is a multiscale estimator" in Section 4.1 and Appendix D.3 of [A] + Figure 1-b of [B], which is precisely an experiment on multivariate normal distributions with specific covariance eigenspectrums that matches the setup in 5.1. Please consider adding these two to the discussions in both papers.

4. Please add the explicit derivation that leads to eq. (19) to the appendix. Especially, the case where $F$ is non-diagonal.

## References

[A] Hamidreza Kamkari, Brendan Leigh Ross, Rasa Hosseinzadeh, Jesse C Cresswell, and Gabriel Loaiza-Ganem. A geometric view of data complexity: Efficient local intrinsic dimension estimation with diffusion models. In Advances in Neural Information Processing Systems, volume 37, 2024.

[B] Tempczyk, Piotr, et al. "A Wiener process perspective on local intrinsic dimension estimation methods." arXiv preprint arXiv:2406.17125 (2024).

**Questions:**

1. What is meant by the ending sentence "in fact, at time $t$ the curvature ..." (L219-L222)?

2. What happens when $t$ is increased for the natural image experiments in Figure 6?

---

### Official Review · Reviewer_gD6B · 2024-11-03

**Soundness:** 1
**Presentation:** 2
**Contribution:** 2
**Rating:** 3
**Confidence:** 3

**Summary:**

This paper studies the eigenvalue spectrum of the Jacobian of (linear) scores of Gaussian distributions supported on low-dimensional subspaces. A particular emphasis is put on memorization, where the empirical score becomes different from the true population score. The times $t$, sample size $N$, and noisy inputs $x$ for where this difference emerges are characterized.

**Strengths:**

I found sections 5.2 and 5.3 enjoyable to read. The effective sample size going from $N$ to $1$ as $t$ decreases clearly evidences the reason for memorization in a quantitative manner, as well as the curse of dimensionality.

**Weaknesses:**

My main issue with the paper is that I don't understand what its contributions are. I do not even know what is being studied: is it the difference between the empirical and the population score? Is the spectrum of the score Jacobian? These two things seem connected in the authors' mind, but I did not see a clear connection. Several results are mentioned in the text and seemingly not used anywhere else. For instance, equation (9) comes from another paper (and in a different setting, as it assumes that $F$ is random, but this is not stated in the text of this paper) and it is not reused. It makes it difficult to see what is the big picture that supposedly emerges from the analysis.

The theoretical results are not clearly stated and do not support the claims made in the text.
- The authors state that "we find that, under some conditions, subspaces of higher variance are lost first due to memorization effects". I did not see where this was shown.
- Similarly, at several places in the text, it is mentioned that the $x$-dependence of the condensation time and other quantities are crucial. However, all numerical experiments are either on Gaussian data, for which the Jacobian is independent of the input, or show the average Jacobian.
- It is stated that "the experimental curves obtained from the empirical score look consistent with the theory". I disagree with this claim. First, the three panels of Figure 4 show qualitatively different behavior. Second, it is not clear to me what predictions the theory was making in the first place, apart from highlighting the presence of gaps at several dimensions (which can be predicted from simple linear algebra).
- In Figure 6, the authors write "The estimated diminsionality tend [sic] to increase with the dataset size, suggesting a phenomenon of geometric memorization." I do not see why an increase of dimensionality would be linked to memorization. On the opposite, shouldn't memorization be characterized by a collapse of dimensionality?

There are several typos. I also suggest using a "sequential" colormap for figures 5 and 6 to enhance readability.

**Questions:**

- If I understand correctly the definition of the manifold $\mathcal M_t$, then it corresponds to the local maxima of $\log p_t$. In the Gaussian case, this corresponds to a single point at the mean of the data distribution for all $t > 0$ (even for singular covariance). How can this illustrate any geometrical properties of the diffusion process?
- Could the authors elaborate on the motivation behind considering the "smoothed" Jacobian?
- Shouldn't $w_\mu(x,t) = p(y^\mu|x) = p(x|y^\mu)/\sum_\nu p(x|y^\nu)$?
- Shouldn't the energy in equation (12) be defined with the opposite sign?
- Line 321, the text mentions "small values of $t$", shouldn't it be "large" instead?
- How does equation (19) derive from equation (18)? I do not see how the variance term remains as the text says it considers the expected Jacobian.

With my current understanding of the paper, I cannot recommend acceptance. I am willing to increase my score if the authors address my points and questions above. In particular, I would like to know what are the contributions of the paper: does it make precise theoretical predictions? How are they verified? Does it evidence new phenomena? What did we learn from this analysis?

---

### Official Review · Reviewer_Xps7 · 2024-11-04

**Soundness:** 2
**Presentation:** 3
**Contribution:** 2
**Rating:** 3
**Confidence:** 4

**Summary:**

Based on the manifold assumption, this paper analyzes the memorization phenomenon in diffusion models. Theoretically, the authors study the memorization effect on an empirical data distribution with its underlying data distribution as a Gaussian distribution. They show that in such case, the specific sampling of the training data points can introduce variances that lead to losing dimensions, i.e., some directions of the tangent space of the manifold are lost. Beyond this theory, they provide empirical results on both synthetic and real world dataset to support the theoretical results.

**Strengths:**

The study is novel and the presentation is good. The observed losing dimensions phenomenon during memorization is very meaningful and opens many interesting research questions.

**Weaknesses:**

The analyze in section 5 is itself very interesting and meaningful, however, I have questions on some of the definitions. Furthermore, I think there exists huge gap between the theoretical results and the practice. i list my questions as follows:

(i) First of all, I have doubt on the definition of the manifold (equation 3). Can you explain why the manifold is defined as the points that has 0 score? This is a very restricted definition since in practice, the $x_t$ along the reverse sampling trajectory never satisfy this assumption unless $t$ is extremely close to 0. Shouldn't we define the manifold as the intermediate noisy distribution $p_t(x)$ ? For the experiment results on natural images, can you clarify whether you are calculating the Jacobians at $x_t$ that have zero score ?


(ii) Without this special manifold assumption, the singular vectors with zero singular values no longer represents the tangent space. Can you clarify whether/how the tangent space interpretation holds when analyzing models trained on natural images ?

(iii) In contrast, the work by Stanczuk does not have this special manifold definition and the dimension of the manifold is calculated in a very different manner rather than just counting the number of zero singular values. I suggest that the authors compare their approach more directly to Stanczuk's method, explaining the key differences and potential implications.


(iv) The definition of memorization is very different from what people care about in practice. In practice, people define memorization as diffusion models generate images that can be found in its training dataset [1]. To be more specific, given a noisy image $x^*_t$ and score function $s(x^*_t,t)$, the denoised version of $x^*_t$ can be written as $x^*_t+\sqrt{t}s(x^*_t,t)$. and memorization happens when $x^*_t+\sqrt{t}s(x^*_t,t)$ is close to training images of the diffusion models. Can the author address how their definition of memorization relates to or differs from the practical definition I've described ?

(v) I think the Jacobians matrix alone cannot fully explain the memorization phenomenon since it doesn't have enough information on the score function $s(x^*_t,t)$. Notice that equation (4) only holds when the deep network is bias free. In general, equation (4) should be written as:
\begin{align}
s(x^*+p,t)\approx s(x^*,t)+J(x^*,t)p.
\end{align}
I think memorization depend more on the property of $s(x^*,t)$ itself, rather than its Jacobian matrix. Can the author explains how their choice of focusing on the bias-free deep network might limit the analysis?

(vi) The analysis on empirical score has huge gap between practice since recent work[2] has shown that denoisers in practice cannot learn such empirical score function.  For this reason, the emprical score function is too ideal and cannot explain the empirical results on natural images (section 6.3).

(vii) The paper claims that memorization causes the model to losing some directions of the tangent spaces. They support this by observing that the time-dependcy gaps in the Jacobian spectrum are lost. However, it should be noted that for diffusion models trained on different number of images, the singular vectors of the Jacobians change significantly as well, which can be observed in the following link
\url{https://github.com/LabForComputationalVision/memorization_generalization_in_diffusion_models/blob/main/notebooks/Demo_UNet_CelebA80x80.ipynb}. Therefore, "losing dimension" is not an accurate description since the subspaces themselves change significantly. The current theory cannot explain this change in subspaces.


[1] Wang, Wenhao, Yifan Sun, Zongxin Yang, Zhengdong Hu, Zhentao Tan, and Yi Yang. "Replication in visual diffusion models: A survey and outlook." arXiv preprint arXiv:2408.00001 (2024).


[2] Zeno, Chen, Greg Ongie, Yaniv Blumenfeld, Nir Weinberger, and Daniel Soudry. "How do minimum-norm shallow denoisers look in function space?." Advances in Neural Information Processing Systems 36 (2024

**Questions:**

I will increase my score if my questions can be addressed.

---

### Note · Authors · 2024-11-27

**Comment:**

The authors thank all the Reviewers for their efforts in reading the manuscript and providing useful comments and questions. We have eventually decided to withdraw our paper, that will be resubmitted in a future venue, certainly improved by your precious suggestions.

**Withdrawal Confirmation:**

I have read and agree with the venue's withdrawal policy on behalf of myself and my co-authors.